# Hierarchical Anchor Graph Learning for Multi-View Clustering

Xingchen Hu [1]   Miao Jia [1]   Jiyuan Liu [1]   Siwei Wang [2]   Ke Liang [1]   Wenjing Yang [1]

## Abstract

Multi-view clustering (MVC) is a fundamental task in heterogeneous data analysis, where anchor-based graph methods are widely adopted for their computational efficiency. However, existing approaches typically utilize static, single-layer anchors, failing to capture the multi-granularity nature of complex data. Drawing inspiration from hierarchical human cognition, we propose a hierarchical anchor graph learning method, termed HAG-MVC, a novel framework that organizes multi-view data as a multi-level pyramid. Unlike conventional one-shot anchor generation methods, HAG-MVC introduces a multi-level co-evolution mechanism, where anchors and graph structures are iteratively refined together to capture semantics from fine-to-coarse granularities. Moreover, HAG-MVC offers a transparent abstraction architecture as an alternative to black-box deep clustering: by maintaining all anchors within the original feature space, it enables explicit inspection of the abstraction process, ensuring inherent interpretability. Extensive experiments on benchmark datasets demonstrate that HAG-MVC consistently outperforms state-of-the-art methods. Beyond MVC, this work provides a scalable and trustworthy paradigm for hierarchical knowledge representation in broad machine learning tasks.

## 1. Introduction

Multi-view clustering (MVC) has emerged as a fundamental paradigm in heterogeneous data analysis (Li et al., 2021; Fang et al., 2023b). With the explosive growth of data scale, anchor-based graph clustering has become attractive due to its computational efficiency (Zhang et al., 2024b; Zhou et al., 2024; Ji et al., 2025). However, most existing approaches rely on a single static anchor set, which typically

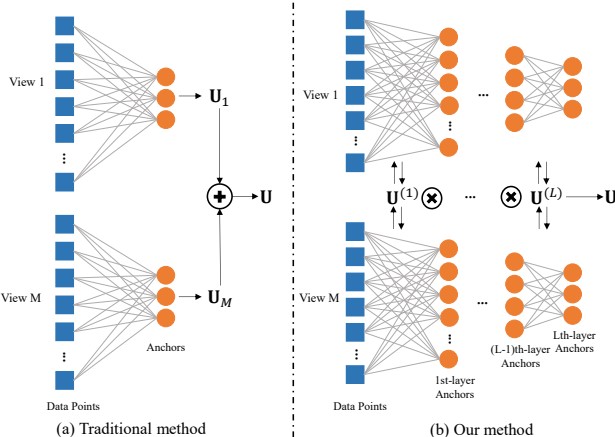

*Figure 1.* Comparison of (a) traditional static single-layer methods and (b) our HAG-MVC. Unlike flat approaches, HAG-MVC employs a multi-level co-evolution mechanism to capture fine-to-coarse semantics within a transparent pyramid structure.

pre-defines a fixed set of anchors through $k$-means or random sampling before the clustering process begins (Yang et al., 2022; 2024). They implicitly assume a fixed representation granularity and overlook the hierarchical abstraction process commonly observed in human cognition.

We argue that this paradigm suffers from two major limitations. First, it decouples anchor selection from graph learning, leading to sub-optimal anchors that may fail to capture the complex, multi-view consensus structure. Second, it assumes a fixed representation granularity. In reality, both human cognition and advanced deep architectures process information through hierarchical abstractions, where data is progressively organized from fine-grained local details to coarse-grained semantic concepts.

To bridge this gap, we propose the Hierarchical Anchor Graph learning for Multi-View Clustering (HAG-MVC) method. It is inspired by a fundamental question: Can we design a framework that possesses the hierarchical abstraction capabilities of deep neural networks while maintaining the transparency and efficiency of anchor-based methods?

The proposed HAG-MVC conceptualizes multi-view data as a multi-level pyramid. As illustrated in Figure 1, instead of a one-shot sampling, we introduce a multi-level co-evolution mechanism. In this architecture, anchors at each layer are

---

[1] National University of Defence Technology, Changsha, China. [2] Academy of Military Sciences, Beijing, China. Correspondence to: Xingchen Hu <xhu4@ualberta.ca>.

*Proceedings of the $43^{rd}$ International Conference on Machine Learning*, Seoul, South Korea. PMLR 306, 2026. Copyright 2026 by the author(s).

not fixed; they are iteratively refined in tandem with the anchor graphs, enabling the model to adaptively "query" information across granularities. This ensures that the anchors at each level are precisely aligned with the semantic abstraction of that layer.

Compared to existing deep clustering methods, HAG-MVC offers a transparent and interpretable alternative. While deep networks often project data into "black-box" hidden spaces, HAG-MVC ensures that anchors at every level reside within the original feature space. This "what-you-see-is-what-you-get" property allows researchers to explicitly inspect the evolution of data structures—from raw samples at the bottom to high-level semantic prototypes at the top. This inherent interpretability is crucial for trustworthy AI in domains where decision-making transparency is as important as performance.

In this work, we shift the multi-view anchor graph learning paradigm from static, single-layered sampling to a dynamic, hierarchical abstraction framework. Our primary contribution is the development of HAG-MVC, a unified architecture that integrates hierarchical anchor learning and graph construction through a multi-level co-evolution mechanism. This design enables the model to capture semantic transitions from fine-grained details to coarse-grained abstractions within a transparent "pyramid" structure. Specifically, we introduce a learnable cross-layer guidance mechanism that propagates structural information across levels while restricting all anchors to the original feature space, thereby ensuring inherent interpretability without sacrificing representational power. Finally, extensive experiments demonstrate that HAG-MVC consistently achieves state-of-the-art performance across multiple benchmarks. Notably, our framework maintains linear computational complexity $O(N)$ with respect to the number of samples, establishing a scalable and trustworthy paradigm for large-scale multi-view representation learning.

## 2. Related Work

To address the scalability issue of multi-view clustering on large-scale datasets, anchor-based multi-view clustering is proposed (Liu et al., 2025), which leverages $c$ anchors to construct an $n \times c$ bipartite graph for clustering, and reduces the time complexity to $\mathcal{O}(nc)$. Traditional anchor-based multi-view clustering methods generally follow a two-step process: constructing individual anchor graphs for each view and integrating them through anchor graph fusion methods (Wang et al., 2022; Zhang et al., 2023; Lu & Feng, 2023). A large number of methods are proposed to explore diverse strategies for anchor graph fusion in multi-view clustering. For example, Kang et al. (Kang et al., 2020) directly assign the same weight to all views, Li et al. (Li et al., 2020) present a parameter-free anchor graph fusion method, and

Zhou et al. (Zhou et al., 2025) employ an auto-weighted allocation strategy to adaptively learn appropriate weight factors for each view. In contrast to the two-step fusion strategy, a number of recent studies attempt to bypass the explicit construction of anchor graphs for individual views. Instead, they directly optimize a unified objective to obtain the fused anchor graph, and reduce intermediate computation and potential inconsistencies across views (Zhang et al., 2024a; Wang et al., 2024). However, the effectiveness of clustering algorithms is highly dependent on the quality of anchor points, making the selection of representative anchors a critical research focus. Early approaches typically employ random sampling from the original dataset to generate anchors (Qiang et al., 2021), which often suffers from instability issues. Therefore, heuristic strategies are developed in several studies, such as $k$-means based methods and feature score based schemes (Yang et al., 2024; 2022). For example, Li et al. (Li et al., 2020) provide a direct alternate sampling method to determine the anchors based on scores. In contrast to these direct anchor selection methods, recent research shifts toward adaptive anchor learning, which simultaneously optimizes latent anchor points and anchor graphs within a unified framework (Liu et al., 2024). For example, Sun et al. (Sun et al., 2021) learn common anchors and the corresponding anchor graph through a projection-based approach. Wen et al. (Wen et al., 2023) propose learning the view-specific anchors and the consistent anchor graph by leveraging local and global structural information.

## 3. Methodology

### 3.1. Proposed Formulation

To address the inherent limitations of single-layer anchor clustering methods in structural modeling, we propose a hierarchical anchor learning method that progressively organizes anchors through multi-layer structural abstraction. In contrast to approaches that rely on a single static sample-anchor bipartite graph, our method models the data structure in a hierarchical manner, thereby enabling richer structural expressiveness and stronger semantic discriminability.

Specifically, the first layer constructs a shared sample-anchor bipartite graph across all views to capture fine-grained local relationships between instances and anchors. In subsequent layers, we learn view-specific anchor sets and establish anchor-anchor graphs that connect the anchors from the $(l-1)$-th layer to those in the $l$-th layer. These inter-layer connections explicitly encode higher-level structural dependencies, allowing information to propagate from bottom to top while preserving the semantic consistency of lower layers. To enhance adaptability, we further introduce learnable transformation mappings between consecutive anchor layers, enabling progressive refinement of anchor embeddings and fusion of multi-scale structural information.

This hierarchical organization offers two significant advantages. First, by progressively optimizing anchor positions through learned structural dependencies, it improves anchor stability, mitigating the instability issues caused by fixed, heuristically sampled anchors. Second, by jointly modeling local instance-level details and global anchor-level abstractions, it enhances representation quality, ensuring alignment between low-level geometric structures and high-level semantic information.

Formally, the data is hierarchically organized into $L$ layers of anchor sets with decreasing cardinality. In the first layer, the data matrix $\mathbf{X}_t$ of each view is approximated by $c_1$ anchors. In each subsequent layer, the anchors from the $(l-1)$-th layer are clustered into $c_l$ higher-level anchors, thereby constructing cross-layer anchor graphs and enabling structural abstraction. This process continues until the final layer is reached, thereby producing a compact and high-level representation of the data structure. The overall optimization objective is defined as follows:

$$
\min_{\mathbf{U}^{(l)}, \mathbf{V}_t^{(l)}} \sum_{t=1}^{M} \left\| \mathbf{X}_t - \mathbf{V}_t^{(1)} \mathbf{U}^{(1)} \right\|_F^2
$$
$$
+ \sum_{l=2}^{L} \sum_{t=1}^{M} \left\| \mathbf{V}_t^{(l-1)} - \mathbf{V}_t^{(l)} \mathbf{U}^{(l)} \right\|_F^2 \qquad (1)
$$
$$
+ \sum_{l=1}^{L} \mu_l \|\mathbf{U}^{(l)}\|_F^2
$$
$$
\text{s.t. } \mathbf{U}^{(l)\top} \mathbf{1}_{c_l} = \mathbf{1}_{c_{l-1}}, \mathbf{U}^{(l)} \geq \mathbf{0}.
$$

Specifically, the first term reconstructs the original input data from the first level anchors, the second term enforces consistency between adjacent anchor layers to facilitate hierarchical abstraction, and the third term regularizes the assignment matrices to prevent overfitting. Here, $\mathbf{X}_t \in \mathbb{R}^{d_t \times n}$ denotes the data matrix of the $t$-th view, which contains $n$ instances with $d_t$-dimensional features, and $M$ denotes the total number of views. $\mathbf{V}_t^{(1)} \in \mathbb{R}^{d_t \times c_1}$ is the first level anchor matrix for the $t$-th view, and $\mathbf{U}^{(1)} \in \mathbb{R}^{c_1 \times n}$ represents the bipartite graph matrix from data to the 1-st hierarchical level anchors. For each layer $l$, $\mathbf{V}_t^{(l)} \in \mathbb{R}^{d_t \times c_l}$ denotes the anchor matrix of the $t$-th view with $c_l$ anchors, where $c_1 > c_2 > \cdots > c_L \geq c$. $\mathbf{U}^{(l)} \in \mathbb{R}^{c_l \times c_{l-1}}$ (for $l \geq 2$) is the anchor-to-anchor graph connecting layers $l - 1$ and $l$. The scalar $\mu_l > 0$ is a regularization parameter that controls the smoothness of the layer-wise graph $\mathbf{U}^{(l)}$, and $\mathbf{1}_k$ denotes a $k$-dimensional column vector with all entries equal to 1.

To facilitate efficient optimization, we further rewrite the objective in Eq. (1) into a trace-based matrix formulation. This equivalent form enables more convenient derivation of update rules for the assignment matrices and is presented as

follows,

$$
\min_{\mathbf{U}^{(l)}, \mathbf{V}_t^{(l)}} \sum_{l=1}^{L} \sum_{t=1}^{M} \text{tr} \left( \mathbf{U}^{(l)\top} \mathbf{V}_t^{(l)\top} \mathbf{V}_t^{(l)} \mathbf{U}^{(l)} \right)
$$
$$
+ \sum_{l=1}^{L-1} \sum_{t=1}^{M} \text{tr} \left( \mathbf{V}_t^{(l)\top} \mathbf{V}_t^{(l)} \right)
$$
$$
- 2 \sum_{t=1}^{M} \text{tr} \left( \mathbf{X}_t^{\top} \mathbf{V}_t^{(1)} \mathbf{U}^{(1)} \right) \qquad (2)
$$
$$
- 2 \sum_{l=2}^{L} \sum_{t=1}^{M} \text{tr} \left( \mathbf{V}_t^{(l-1)\top} \mathbf{V}_t^{(l)} \mathbf{U}^{(l)} \right)
$$
$$
+ \sum_{l=1}^{L} \mu_l \text{tr} \left( \mathbf{U}^{(l)\top} \mathbf{U}^{(l)} \right)
$$
$$
\text{s.t. } \mathbf{U}^{(l)} \geq 0, \mathbf{U}^{(l)\top} \mathbf{1} = \mathbf{1}.
$$

Then, we iteratively update the anchor graphs and anchor matrices until convergence. As a result, we produce the final anchor graph $\mathbf{U} = \prod_{l=1}^{L} \mathbf{U}^{(l)\top}$. As established in (Wang et al., 2024; Kang et al., 2020), the left singular vectors of the anchor graph $\mathbf{U}$ are equivalent to eigenvectors of the full similarity graph $\mathbf{S} = \mathbf{U}\mathbf{U}^{\top}$. Based on this, we perform singular value decomposition (SVD) on $\mathbf{U}$ to obtain $\mathbf{H}$, followed by applying $k$-means on $\mathbf{H}$ to derive the final clustering results.

### 3.2. Optimization Algorithm

To solve the objective in Eq. (2), we adopt an alternating optimization strategy. Given that the objective is not jointly convex with respect to all variables $\{\mathbf{U}^{(l)}\}_{l=1}^{L}$ and $\{\mathbf{V}_t^{(l)}\}_{l=1}^{L}$, we update each variable iteratively while fixing the others. The specific procedure is derived from the following detailed steps, and the complete solving process is summarized in Algorithm 1.

**Update $\mathbf{U}^{(1)}$.** Fixing $\{\mathbf{U}^{(l)}\}_{l=2}^{L}$ and $\{\mathbf{V}_t^{(l)}\}_{l=1}^{L}$, and ignoring the terms irrelevant to $\mathbf{U}^{(1)}$, the optimization problem in Eq. (2) with respect to $\mathbf{U}^{(1)}$ can be reformulated as

$$
\min_{\mathbf{U}^{(1)}} \sum_{t=1}^{M} \text{tr} \left( \mathbf{U}^{(1)\top} \mathbf{V}_t^{(1)\top} \mathbf{V}_t^{(1)} \mathbf{U}^{(1)} \right)
$$
$$
- 2 \sum_{t=1}^{M} \text{tr} \left( \mathbf{X}_t^{\top} \mathbf{V}_t^{(1)} \mathbf{U}^{(1)} \right) + \mu_1 \text{tr} \left( \mathbf{U}^{(1)\top} \mathbf{U}^{(1)} \right)
$$
$$
\text{s.t. } \mathbf{U}^{(1)} \geq 0, \mathbf{U}^{(1)\top} \mathbf{1} = \mathbf{1}.
$$
$$
(3)
$$

Since the constraints on $\mathbf{U}^{(1)}$ are imposed column-wise, the optimization process of $\mathbf{U}^{(1)}$ in Eq. (3) can be easily

expressed as the following quadratic programming (QP) problem,

$$\min_{\mathbf{U}_{[:,k]}^{(1)}} \frac{1}{2} \mathbf{U}_{[:,k]}^{(1)\top} \mathbf{Q}^{(1)} \mathbf{U}_{[:,k]}^{(1)} + \mathbf{p}_k^{(1)\top} \mathbf{U}_{[:,k]}^{(1)} \tag{4}$$

$$\text{s.t. } \mathbf{U}_{[:,k]}^{(1)\top} \mathbf{1} = 1, \mathbf{U}_{[:,k]}^{(1)} \geq 0,$$

where $\mathbf{Q}^{(1)} = \sum_{t=1}^{M} \left( \mathbf{V}_t^{(1)\top} \mathbf{V}_t^{(1)} \right) + \mu_1 \mathbf{I}$, $\mathbf{p}_k^{(1)} = -\sum_{t=1}^{M} \left( \mathbf{X}_{t[:,k]}^{\top} \mathbf{V}_t^{(1)} \right)^{\top}$, and $\mathbf{U}_{[:,k]}^{(1)}$ denotes the $k$-th column of $\mathbf{U}^{(1)}$. By independently solving the column-wise QP problem in Eq. (4), we obtain all column vectors and concatenate them to form the updated $\mathbf{U}^{(1)}$.

**Update $\mathbf{V}_t^{(1)}$.** Fixing $\{\mathbf{U}^{(l)}\}_{l=1}^{L}$ and $\{\mathbf{V}_t^{(l)}\}_{l=2}^{L}$, and ignoring the unrelated terms, the optimization problem in Eq. (2) for updating $\mathbf{V}_t^{(1)}$ can be written as follows,

$$\min_{\mathbf{V}_t^{(1)}} \sum_{t=1}^{M} \text{tr} \left( \mathbf{V}_t^{(1)} \mathbf{U}^{(1)} \mathbf{U}^{(1)\top} \mathbf{V}_t^{(1)\top} \right) + \sum_{t=1}^{M} \text{tr} \left( \mathbf{V}_t^{(1)} \mathbf{V}_t^{(1)\top} \right)$$

$$- 2 \sum_{t=1}^{M} \text{tr} \left( \mathbf{U}^{(1)} \mathbf{X}_t^{\top} \mathbf{V}_t^{(1)} \right) - 2 \sum_{t=1}^{M} \text{tr} \left( \mathbf{U}^{(2)\top} \mathbf{V}_t^{(2)\top} \mathbf{V}_t^{(1)} \right) \tag{5}$$

Since the optimization of each $\mathbf{V}_t^{(1)}$ is independent of the other views, we take the derivative of the objective function with respect to $\mathbf{V}_t^{(1)}$ and set it to zero. The solution can be computed as

$$\mathbf{V}_t^{(1)} = \left( \mathbf{X}_t \mathbf{U}^{(1)\top} + \mathbf{V}_t^{(2)} \mathbf{U}^{(2)} \right) \left( \mathbf{U}^{(1)} \mathbf{U}^{(1)\top} + \mathbf{I} \right)^{-1}. \tag{6}$$

**Update $\{\mathbf{U}^{(l)}\}_{l=2}^{L}$.** Fixing $\mathbf{U}^{(1)}$ and $\{\mathbf{V}_t^{(l)}\}_{l=1}^{L}$, the optimization problem in Eq. (2) with respect to $\{\mathbf{U}^{(l)}\}_{l=2}^{L}$ can be written as

$$\min_{\mathbf{U}^{(l)}} \sum_{l=2}^{L} \sum_{t=1}^{M} \text{tr} \left( \mathbf{U}^{(l)\top} \mathbf{V}_t^{(l)\top} \mathbf{V}_t^{(l)} \mathbf{U}^{(l)} \right)$$

$$- 2 \sum_{l=2}^{L} \sum_{t=1}^{M} \text{tr} \left( \mathbf{V}_t^{(l-1)\top} \mathbf{V}_t^{(l)} \mathbf{U}^{(l)} \right) \tag{7}$$

$$+ \sum_{l=2}^{L} \mu_l \text{tr} \left( \mathbf{U}^{(l)\top} \mathbf{U}^{(l)} \right)$$

$$\text{s.t. } \mathbf{U}^{(l)} \geq 0, \mathbf{U}^{(l)\top} \mathbf{1} = \mathbf{1}.$$

The optimization process of $\mathbf{U}^{(l)}$ can be expressed as the following QP problem,

$$\min_{\mathbf{U}_{[:,k]}^{(l)}} \frac{1}{2} \mathbf{U}_{[:,k]}^{(l)\top} \mathbf{Q}^{(l)} \mathbf{U}_{[:,k]}^{(l)} + \mathbf{p}_k^{(l)\top} \mathbf{U}_{[:,k]}^{(l)} \tag{8}$$

$$\text{s.t. } \mathbf{U}_{[:,k]}^{(l)\top} \mathbf{1} = 1, \mathbf{U}_{[:,k]}^{(l)} \geq 0,$$

**Algorithm 1** HAG-MVC

1: **Input:** multi-view datasets $\{\mathbf{X}_t\}_{t=1}^{M}$, number of clusters $k$, number of layers $L$, number of anchors $\{c_l\}_{l=1}^{L}$, parameter $\{\mu_l\}_{l=1}^{L}$, maximal iteration $\Gamma$.
2: **Initialize** $\{\mathbf{V}_t^{(l)}\}_{t=1,l=1}^{M,L}$ and $\{\mathbf{U}^{(l)}\}_{l=1}^{L}$.
3: **while** not converged **and** iteration $< \Gamma$ **do**
4:    **for** $l = 1$ **to** $L$ **do**
5:       Update $\mathbf{U}^{(l)}$ by solving (4), (8).
6:       Update $\{\mathbf{V}_t^{(l)}\}_{t=1}^{M}$ by solving (6), (9), (10).
7:    **end for**
8: **end while**
9: **Output:** Predicted labels $\widehat{\mathbf{Y}}$.

where $\mathbf{Q}^{(l)} = \sum_{t=1}^{M} \left( \mathbf{V}_t^{(l)\top} \mathbf{V}_t^{(l)} \right) + \mu_l \mathbf{I}$, and $\mathbf{p}_k^{(l)} = -\sum_{t=1}^{M} \left( \mathbf{V}_{t[:,k]}^{(l-1)\top} \mathbf{V}_t^{(l)} \right)^{\top}$.

**Update $\left\{ \mathbf{V}_t^{(l)} \right\}_{l=2}^{L-1}$.** Fixing the other variables and ignoring the unrelated terms, the solution is obtained by taking the derivative with respect to $\mathbf{V}_t^{(l)}$ and setting it to zero,

$$\mathbf{V}_t^{(l)} = \mathbf{B}_t^{(l)} \left( \mathbf{U}^{(l)} \mathbf{U}^{(l)\top} + \mathbf{I} \right)^{-1}, \tag{9}$$

where $\mathbf{B}_t^{(l)} = \mathbf{V}_t^{(l-1)} \mathbf{U}^{(l)\top} + \mathbf{V}_t^{(l+1)} \mathbf{U}^{(l+1)}$.

**Update $\mathbf{V}_t^{(L)}$.** Fixing other variables and ignoring the unrelated terms, the solution can be computed as

$$\mathbf{V}_t^{(L)} = \mathbf{V}_t^{(L-1)} \mathbf{U}^{(L)\top} \left( \mathbf{U}^{(L)} \mathbf{U}^{(L)\top} \right)^{-1}. \tag{10}$$

### 3.3. Discussion of the Algorithm

#### 3.3.1. CONVERGENCE

The overall objective function in Eq. (2) is not jointly convex with respect to all variables $\{\mathbf{U}^{(l)}\}_{l=1}^{L}$ and $\{\mathbf{V}_t^{(l)}\}_{l=1}^{L}$. To address this, we adopt an alternate optimization strategy that updates each variable while fixing the others. Each subproblem admits either a closed-form solution or can be efficiently solved by quadratic programming. Let $\mathcal{J}^{(i)}$ denote the objective function value at the $i$-th iteration. Then the update process satisfies $\mathcal{J}^{(i)} \geq \mathcal{J}^{(i+1)}$, indicating that the objective value monotonically decreases after each iteration. Since the objective function is lower bounded by zero, the proposed algorithm is guaranteed to converge to a local minimum.

#### 3.3.2. COMPLEXITY ANALYSIS

**Time Complexity.** The time consumption of our hierarchical anchor learning framework mainly arises from

Table 1. A brief summary of multi-view datasets.

| Datasets | Samples | Clusters | Dimensions |
|---|---|---|---|
| MSRC | 210 | 7 | 24/576/512/256/254 |
| Dermatology | 358 | 6 | 12/22 |
| ForestTypes | 523 | 4 | 9/9/9 |
| PIE | 1629 | 68 | 1024/1024/1024 |
| MFeat | 2000 | 10 | 76/240 |
| HandWritten | 2000 | 10 | 240/76/216/47/64/6 |
| BDGP | 2500 | 5 | 1000/500/250 |
| CCV | 6773 | 20 | 20/20/20 |
| COIL100 | 7200 | 100 | 30/99/30 |
| VGGFace2 | 16936 | 50 | 944/576/512/640 |
| Reuters | 18758 | 6 | 21513/24892/34251/15506/11547 |
| YoutubeFace | 101499 | 31 | 64/512/64/647/838 |

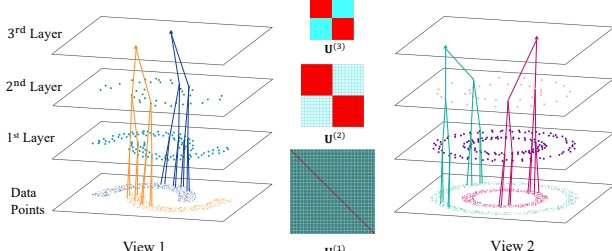

Figure 2. Visualization of the hierarchical clustering process on synthetic data. The figure displays the distribution of learned anchors at the first and second layers. The proposed HAG-MVC effectively learns local landmarks at the first layer and aggregates them into semantic prototypes at the second layer, providing an interpretable path for clustering decisions and error diagnosis.

the iterative updates of the bipartite assignment matrices $\{\mathbf{U}^{(l)}\}_{l=1}^L$ and anchor representations $\{\mathbf{V}_t^{(l)}\}_{l=1}^L$. Below, we analyze the time complexity of each update step. For updating the sample-to-anchor graph $\mathbf{U}^{(1)} \in \mathbb{R}^{c_1 \times n}$, the complexity consists of computing matrix products and solving $n$ QP subproblems. The overall complexity is $\mathcal{O}(dnc_1 + dc_1^2 + nc_1^3)$, where $d = \sum_{t=1}^M d_t$ denotes the total feature dimension across all views. Similarly, for updating the cross-layer anchor graph $\mathbf{U}^{(l)} \in \mathbb{R}^{c_l \times c_{l-1}}$ with $l \geq 2$, the time complexity is $\mathcal{O}(dc_l^2 + dc_l c_{l-1} + c_{l-1} c_l^3)$. For updating the anchor matrix $\mathbf{V}_t^{(1)} \in \mathbb{R}^{d_t \times c_1}$, the time complexity is $\mathcal{O}(dnc_1 + Mc_1^3)$. For updating the intermediate anchor representations $\{\mathbf{V}_t^{(l)}\}_{l=2}^L$, the time complexity is $\mathcal{O}(dc_{l-1}c_l + Mc_l^3)$. Overall, since the number of anchors $c_l \ll n$, the proposed method maintains favorable scalability, and the total time complexity grows linearly with respect to the number of data points $n$, i.e., $\mathcal{O}(n)$, which enables our model to scale efficiently to large datasets.

**Space Complexity.** The overall space complexity of our method is primarily attributed to storing the hierarchical anchor matrices $\{\mathbf{V}_t^{(l)}\}_{l=1}^L$, the cross-layer anchor graph $\{\mathbf{U}^{(l)}\}_{l=2}^L$, and the sample-to-anchor graph $\mathbf{U}^{(1)} \in \mathbb{R}^{c_1 \times n}$. Considering $M$ views and $L$ hierarchical levels, the total space cost is $\mathcal{O}(c_1 n + \sum_{l=2}^L c_l c_{l-1} + d \sum_{l=1}^L c_l)$, where $d = \sum_{t=1}^M d_t$. Since the number of anchors $c_l \ll n$, the space complexity increases linearly with respect to $n$, thus ensuring the scalability of our model on large-scale multi-view datasets.

## 4. Experimental Studies

### 4.1. Visualization on Synthetic Data

Before the quantitative benchmarks, we conduct an illustrative experiment on a synthetic multi-view dataset to visually elucidate the hierarchical clustering mechanism. As depicted in Figure 2, HAG-MVC effectively captures local geometric structures via the first-layer anchors and progressively abstracts them into high-level semantic prototypes at

the second layer. This "fine-to-coarse" evolution transparently reveals how the model distills consistent cluster centers from low-level features, differentiating it from opaque "black-box" deep clustering methods.

The proposed framework offers a transparent perspective for error diagnosis. By examining the bipartite connectivity, we can explicitly trace the decision path for any specific sample. For misclassified points, this transparency allows us to pinpoint whether the error stems from ambiguous local boundaries in the first layer or incorrect semantic aggregation in deeper layers. This capability not only ensures the interpretability of the clustering process but also provides valuable insights for diagnosing method performance.

### 4.2. Experimental Settings

**Multi-View Datasets.** In our experiments, we conduct extensive evaluations on twelve widely-used multi-view benchmark datasets, including MSRC, Dermatology (Derma), ForestTypes (Forest), PIE, MFeat, HandWritten (HW), BDGP, CCV, COIL100, VGGFace2 (VGGF2), Reuters, and YoutubeFace (YTF). Detailed statistics of these datasets are summarized in Table 1.

**Compared Algorithms.** To evaluate the effectiveness of the proposed HAG-MVC, we compare it against the following eleven anchor-based multi-view clustering baselines. 1) *LMVSC* (Kang et al., 2020) employs anchor-based graph learning for linear-time multi-view clustering. 2) *SMVSC* (Sun et al., 2021) integrates anchor learning and affinity construction into a unified optimization. 3) *OMSC* (Chen et al., 2022) integrates the consensus anchor representation, consensus graph, and final clusterings into a single coherent framework. 4) *FMVACC* (Wang et al., 2022) addresses the anchor-unaligned problem through flexible anchor graph fusion. 5) *FastMICE* (Huang et al., 2023) enhances clustering robustness through random view

groups and hybrid fusion. 6) *UDBGL* (Fang et al., 2023a) learns unified discrete bipartite graphs with a Laplacian rank constraint. 7) *Orth-NTF* (Li et al., 2023) leverages orthogonal tensor factorization and Schatten $p$-norm regularization to capture high-order inter-view correlations. 8) *MVSC-HFD* (Ou et al., 2024) projects data into a shared subspace using hierarchical projection matrices. 9) *MV-CAGAF* (Wang et al., 2025) resolves the cross-view anchor misalignment issue via structural representation matching. 10) *3AMVC* (Ma et al., 2024) automatically identifies and aligns high-quality anchors across views. 11) *HBG-MVSC* (Zhou et al., 2025) constructs hierarchical bipartite graphs with auto-weighted fusion for clustering.

**Experimental Details.** The hyperparameters of all baselines are tuned through grid search within the ranges suggested in the original papers to ensure optimal performance. For methods involving $k$-means, we select the best result from 50 runs to mitigate randomness. For our method, the initialization scheme adopted in this work is based on $k$-means. In practice, we traverse the number of hierarchical layers $L$ from 1 to 4 to explore the effect of depth, and subsequent experiments demonstrate the rationale for selecting at most four layers. For each layer $l$, the number of anchors is selected from a predefined set $\{10k, 9k, \ldots, k\}$, where $k$ denotes the ground-truth number of clusters. This configuration ensures that deeper hierarchies correspond to progressively coarser anchor representations. The regularization parameter $\mu_l$ for each layer is uniformly set to $1/L$, so that the regularization terms are equally weighted across layers. Apart from the anchor configurations, the model does not involve any additional tunable hyperparameters. To be specific, the experiments in this paper are conducted on a computer with a 1.40 GHz Intel Core Ultra 7 CPU and 32GB RAM.

### 4.3. Comparisons with State-of-the-Art Methods

Table 2 presents the clustering performance of our method compared with eleven state-of-the-art methods. The best and second best results in all methods are represented by bold value and underline value, respectively. According to the results, we have the following observations:

- HAG-MVC shows clear advantages over other multi-view clustering baselines. Especially in terms of ACC, our proposed algorithm outperforms all the compared methods across all datasets. Our method improves 1.87%, 2.78%, 5.02%, 9.60%, 7.86%, 1.06%, 1.58%, 10.00%, 0.13%, 15.20%, 15.24% and 11.03% over the second best method on twelve datasets, respectively. These results clearly demonstrate the effectiveness and superiority of our method.

- Compared with several classic large-scale multi-view

clustering methods that adopt bipartite graph construction, anchor matching correspondences, or projection strategies (UDBGL, FMVACC and SMVSC), our HAG-MVC demonstrates superior performance. This validates the effectiveness of our multi-level anchor learning strategy guided by hierarchical anchor graph construction.

- On some datasets like MFeat, HW and BDGP, our method achieves clear improvements over several dynamic anchor learning approaches, such as MV-CAGAF and MVSC-HFD. This is because our method incorporates structural constraints that regularize the anchor assignment process, leading to more robust and semantically consistent similarity graphs.

### 4.4. Ablation Study

**Ablation study on fixed and learned anchors.** In our proposed method, the anchors of each view and the consensus anchor graph are jointly learned, avoiding late fusion. In contrast to hierarchical approaches that adopt a fixed-anchor strategy, we treat the anchor sets $\{\mathbf{V}_t^{(l)}\}_{l=1}^{L}$ as learnable variables to better capture cross-view structural consistency. As shown in Table 3, the learned-anchor strategy consistently outperforms the fixed-anchor setting across twelve datasets in ACC, NMI, Purity, and F-score. For example, ACC on PIE improves from 0.2487 to 0.6028, and on Reuters from 0.3801 to 0.6035. These results confirm that adaptively updating anchors better captures discriminative structures across views and abstraction levels, leading to more robust and semantically consistent similarity graphs.

**Ablation study on hierarchical anchor strategies.** To verify the importance of the proposed hierarchical anchor design, we compare HAG-MVC with a variant that removes the hierarchical anchor term. As shown in Table 4, incorporating hierarchical anchoring consistently improves clustering performance across all metrics and datasets. The gains are particularly evident on MSRC, MFeat, and HW, where ACC and Purity increase by over 3%, demonstrating that jointly learning multi-layer anchors and the consensus anchor graph effectively captures cross-view structural consistency. These results confirm that the hierarchical anchoring strategy provides a stable and substantial advantage over its non-hierarchical counterpart.

### 4.5. Parameter Sensitivity

To further examine the impact of the hierarchical anchor design on clustering performance, we conducted two sets of controlled experiments with different anchor configurations and hierarchy depths.

*Table 2.* Clustering performance comparison on twelve datasets. 'N/A' indicates execution failure due to high complexity.

| Dataset | LMVSC (AAAI'20) | SMVSC (MM'21) | OMSC (KDD'22) | FMVACC (NeurIPS'22) | FastMICE (TKDE'23) | UDBGL (TNNLS'23) | Orth-NTF (NeurIPS'23) | MVSC-HFD (IF'24) | 3AMVC (MM'24) | MV-CAGAF (TKDE'25) | HBG-MVSC (IF'25) | Ours |
|---|---|---|---|---|---|---|---|---|---|---|---|---|
| **ACC** | | | | | | | | | | | | |
| MSRC | 0.3429 | 0.8190 | 0.7952 | 0.7171 | 0.8381 | 0.7905 | 0.8143 | 0.6571 | 0.4920 | 0.7143 | 0.7381 | **0.8538** |
| Derma | 0.7877 | 0.8408 | 0.8101 | 0.8439 | 0.8212 | 0.7039 | 0.4637 | 0.8771 | 0.6522 | 0.8623 | 0.8296 | **0.9015** |
| Forest | 0.7935 | 0.7247 | 0.6711 | 0.7787 | 0.6979 | 0.7839 | 0.5717 | 0.7170 | 0.7186 | 0.7187 | 0.6979 | **0.8333** |
| PIE | 0.2400 | 0.1627 | 0.1529 | 0.5391 | 0.2185 | 0.1977 | 0.5500 | 0.1627 | 0.4499 | 0.2793 | 0.1522 | **0.6028** |
| MFeat | 0.8450 | 0.7735 | 0.7465 | 0.8498 | 0.8640 | 0.8110 | 0.7945 | 0.8105 | 0.8377 | 0.7764 | 0.8145 | **0.9319** |
| HW | 0.8540 | 0.8205 | 0.8065 | 0.9018 | 0.7660 | 0.7220 | 0.9220 | 0.6545 | 0.7768 | 0.7358 | 0.8705 | **0.9318** |
| BDGP | 0.4552 | 0.5236 | 0.4928 | 0.5374 | 0.4952 | 0.3928 | 0.4366 | 0.4192 | 0.4533 | 0.5625 | 0.4304 | **0.5714** |
| CCV | 0.2073 | 0.2210 | 0.2134 | 0.1919 | 0.2126 | 0.1813 | 0.1540 | 0.2048 | 0.1747 | 0.1692 | 0.1392 | **0.2431** |
| COIL100 | 0.6282 | 0.7246 | 0.7196 | 0.6261 | 0.7801 | 0.6388 | 0.3983 | 0.4399 | 0.6223 | 0.2914 | 0.4251 | **0.7811** |
| VGGF2 | 0.1017 | 0.0933 | 0.0948 | 0.1092 | 0.1027 | 0.0838 | 0.0905 | 0.1080 | 0.1054 | 0.0688 | 0.0671 | **0.1258** |
| Reuters | 0.4279 | N/A | N/A | 0.5237 | 0.4582 | N/A | N/A | N/A | 0.3633 | 0.4623 | 0.2838 | **0.6035** |
| YTF | 0.1460 | N/A | 0.2350 | 0.1294 | 0.2173 | N/A | N/A | N/A | 0.1115 | 0.2332 | 0.2574 | **0.2858** |
| **NMI** | | | | | | | | | | | | |
| MSRC | 0.2465 | 0.7176 | 0.6911 | 0.6232 | 0.7483 | 0.7120 | 0.7695 | 0.5346 | 0.4170 | 0.5903 | 0.6780 | **0.7714** |
| Derma | 0.7758 | 0.7985 | 0.7703 | 0.7627 | 0.7634 | 0.7495 | 0.2663 | 0.8042 | 0.6253 | 0.8450 | 0.7264 | **0.8613** |
| Forest | 0.5437 | 0.4407 | 0.4023 | 0.5397 | 0.4596 | 0.5476 | 0.2820 | 0.4219 | 0.4771 | 0.3970 | 0.4108 | **0.6036** |
| PIE | 0.5299 | 0.3021 | 0.2240 | 0.7636 | 0.5343 | 0.4720 | 0.6845 | 0.2907 | 0.7202 | 0.5631 | 0.3871 | **0.8048** |
| MFeat | 0.8119 | 0.7313 | 0.7310 | 0.8155 | 0.8597 | 0.7851 | 0.8333 | 0.7262 | 0.7987 | 0.7283 | 0.8416 | **0.8636** |
| HW | 0.8081 | 0.7886 | 0.7726 | 0.8625 | 0.8469 | 0.7329 | **0.8932** | 0.6764 | 0.7391 | 0.6969 | 0.8606 | 0.8801 |
| BDGP | 0.2528 | 0.2802 | 0.2571 | 0.3597 | 0.3108 | 0.1504 | 0.2416 | 0.1535 | 0.2133 | **0.4054** | 0.1384 | 0.3618 |
| CCV | 0.1681 | 0.1694 | 0.1686 | 0.1505 | 0.1653 | 0.1455 | 0.0902 | 0.1528 | 0.1293 | 0.1413 | 0.1138 | **0.1977** |
| COIL100 | 0.8458 | 0.8309 | 0.8505 | 0.8256 | **0.9262** | 0.8017 | 0.6786 | 0.7258 | 0.8212 | 0.5338 | 0.7332 | 0.9117 |
| VGGF2 | 0.1219 | 0.1153 | 0.1302 | 0.1169 | 0.1219 | 0.1008 | 0.0901 | 0.1343 | 0.1194 | 0.0898 | 0.0825 | **0.1552** |
| Reuters | 0.2516 | N/A | N/A | 0.3453 | 0.3020 | N/A | N/A | N/A | 0.2271 | 0.2981 | 0.2290 | **0.3550** |
| YTF | 0.1345 | N/A | 0.2269 | 0.0946 | 0.1948 | N/A | N/A | N/A | 0.0826 | 0.0577 | 0.0127 | **0.2436** |
| **Purity** | | | | | | | | | | | | |
| MSRC | 0.3810 | 0.8190 | 0.7952 | 0.7368 | 0.8381 | 0.7905 | 0.8143 | 0.6714 | 0.5144 | 0.7190 | 0.8190 | **0.8556** |
| Derma | 0.8911 | 0.8408 | 0.8464 | 0.8538 | 0.8436 | 0.8408 | 0.5028 | 0.8771 | 0.7151 | 0.9014 | 0.8296 | **0.9015** |
| Forest | 0.7935 | 0.7247 | 0.6711 | 0.7787 | 0.6979 | 0.7839 | 0.6119 | 0.7170 | 0.7437 | 0.7187 | 0.6979 | **0.8333** |
| PIE | 0.3235 | 0.1700 | 0.1572 | 0.5711 | 0.2584 | 0.2535 | 0.5537 | 0.1700 | 0.4790 | 0.3039 | 0.1946 | **0.6288** |
| MFeat | 0.8450 | 0.7735 | 0.7700 | 0.8604 | 0.8675 | 0.8700 | 0.8305 | 0.8105 | 0.8491 | 0.7791 | 0.9120 | **0.9319** |
| HW | 0.8540 | 0.8205 | 0.8065 | 0.9078 | 0.8655 | 0.8160 | 0.9220 | 0.6570 | 0.7835 | 0.7654 | 0.8705 | **0.9347** |
| BDGP | 0.5064 | 0.5320 | 0.5112 | 0.5530 | 0.5560 | 0.4204 | 0.4524 | 0.4192 | 0.4742 | **0.6062** | 0.4304 | 0.5758 |
| CCV | 0.2337 | 0.2442 | 0.2390 | 0.2308 | 0.2429 | 0.2491 | 0.1880 | 0.2379 | 0.2181 | 0.2108 | 0.2541 | **0.2687** |
| COIL100 | 0.7365 | 0.7336 | 0.7244 | 0.6477 | **0.8729** | 0.7644 | 0.4242 | 0.4536 | 0.6500 | 0.3089 | 0.4399 | 0.8150 |
| VGGF2 | 0.1124 | 0.0976 | 0.0994 | 0.1194 | 0.1251 | 0.1071 | 0.1023 | 0.1127 | 0.1115 | 0.0733 | **0.1793** | 0.1348 |
| Reuters | 0.6002 | N/A | N/A | 0.6178 | 0.5545 | N/A | N/A | N/A | 0.4437 | 0.5411 | 0.3859 | **0.6316** |
| YTF | 0.1944 | N/A | 0.3234 | 0.2760 | 0.2455 | N/A | N/A | N/A | 0.2715 | 0.2765 | 0.3441 | **0.3669** |
| **F-score** | | | | | | | | | | | | |
| MSRC | 0.2474 | 0.6988 | 0.6891 | 0.6057 | 0.7372 | 0.6904 | 0.7234 | 0.5224 | 0.3356 | 0.5590 | 0.6296 | **0.7549** |
| Derma | 0.7762 | 0.8371 | 0.7916 | 0.7801 | 0.8198 | 0.6079 | 0.3557 | 0.8198 | 0.5799 | 0.8363 | 0.7848 | **0.8632** |
| Forest | 0.6572 | 0.5673 | 0.5200 | 0.6295 | 0.5567 | 0.6409 | 0.4341 | 0.5578 | 0.5981 | 0.5536 | 0.5569 | **0.7073** |
| PIE | 0.1320 | 0.0351 | 0.0328 | 0.3974 | 0.1340 | 0.0898 | 0.4242 | 0.0358 | 0.3441 | 0.1604 | 0.0596 | **0.5031** |
| MFeat | 0.7744 | 0.6791 | 0.6665 | 0.7772 | 0.8212 | 0.7389 | 0.7640 | 0.6832 | 0.7595 | 0.6822 | 0.7705 | **0.8714** |
| HW | 0.7757 | 0.7525 | 0.7321 | 0.8484 | 0.7664 | 0.6614 | 0.8698 | 0.5696 | 0.6534 | 0.6539 | 0.8285 | **0.8793** |
| BDGP | 0.3741 | 0.3848 | 0.3574 | 0.4067 | 0.4154 | 0.2947 | 0.3282 | 0.3254 | 0.3337 | **0.4675** | 0.3291 | 0.4217 |
| CCV | 0.1160 | 0.1236 | 0.1318 | 0.1143 | 0.1251 | 0.1111 | 0.0859 | 0.1215 | 0.1042 | 0.1106 | 0.0969 | **0.1393** |
| COIL100 | 0.5793 | 0.5139 | 0.6433 | 0.5697 | **0.7584** | 0.3481 | 0.2854 | 0.3724 | 0.5783 | 0.2070 | 0.3709 | 0.7518 |
| VGGF2 | 0.0468 | 0.0551 | 0.0557 | 0.0498 | 0.0506 | 0.0418 | 0.0386 | 0.0563 | 0.0540 | 0.0458 | 0.0425 | **0.0634** |
| Reuters | 0.3880 | N/A | N/A | 0.4385 | 0.4086 | N/A | N/A | N/A | 0.3363 | 0.3999 | 0.3493 | **0.4743** |
| YTF | 0.0774 | N/A | 0.1199 | 0.0656 | 0.0925 | N/A | N/A | N/A | 0.0612 | 0.1551 | **0.1633** | 0.1234 |

*Table 3.* Ablation study on learned anchor strategy.

| Metric | Strategy | Datasets | | | | | | | | | | | |
|---|---|---|---|---|---|---|---|---|---|---|---|---|---|
| | | MSRC | Derma | Forest | PIE | MFeat | HW | BDGP | CCV | COIL100 | VGGF2 | Reuters | YTF |
| ACC | Fixed | 0.7851 | 0.6853 | 0.5241 | 0.2487 | 0.8770 | 0.8379 | 0.4764 | 0.1630 | 0.7139 | 0.0840 | 0.3801 | 0.2381 |
| | Learned | **0.8538** | **0.9015** | **0.8333** | **0.6028** | **0.9319** | **0.9318** | **0.5714** | **0.2431** | **0.7811** | **0.1258** | **0.6035** | **0.2858** |
| NMI | Fixed | 0.6708 | 0.6949 | 0.2977 | 0.5475 | 0.8123 | 0.7757 | 0.2182 | 0.1171 | 0.8789 | 0.1012 | 0.1324 | 0.2016 |
| | Learned | **0.7714** | **0.8613** | **0.6036** | **0.8048** | **0.8636** | **0.8801** | **0.3618** | **0.1977** | **0.9117** | **0.1552** | **0.3550** | **0.2436** |
| Purity | Fixed | 0.7884 | 0.7882 | 0.5943 | 0.2682 | 0.8824 | 0.8844 | 0.4785 | 0.2119 | 0.7523 | 0.0936 | 0.4577 | 0.3329 |
| | Learned | **0.8556** | **0.9015** | **0.8333** | **0.6288** | **0.9319** | **0.9347** | **0.5758** | **0.2687** | **0.8150** | **0.1348** | **0.6316** | **0.3669** |
| F-score | Fixed | 0.6592 | 0.6655 | 0.4423 | 0.1451 | 0.7987 | 0.7481 | 0.3512 | 0.0984 | 0.6757 | 0.0391 | 0.3100 | 0.1061 |
| | Learned | **0.7549** | **0.8632** | **0.7073** | **0.5031** | **0.8714** | **0.8793** | **0.4217** | **0.1393** | **0.7518** | **0.0634** | **0.4743** | **0.1234** |

*Table 4.* Ablation study of the hierarchical anchor learning strategy.

| Metric | Strategy | Datasets | | | | | | | | | | | |
|---|---|---|---|---|---|---|---|---|---|---|---|---|---|
| | | MSRC | Derma | Forest | PIE | MFeat | HW | BDGP | CCV | COIL100 | VGGF2 | Reuters | YTF |
| ACC | Single-Layer | 0.7656 | 0.7804 | 0.7007 | 0.5989 | 0.8930 | 0.8918 | 0.4678 | 0.2091 | 0.6646 | 0.1158 | 0.4653 | 0.2642 |
| | Hierarchical | **0.8538** | **0.9015** | **0.8333** | **0.6028** | **0.9319** | **0.9318** | **0.5714** | **0.2431** | **0.7811** | **0.1258** | **0.6035** | **0.2858** |
| NMI | Single-Layer | 0.6490 | 0.7499 | 0.3891 | 0.8033 | 0.8312 | 0.8613 | 0.2687 | 0.1856 | 0.8313 | 0.1370 | 0.2773 | 0.2394 |
| | Hierarchical | **0.7714** | **0.8613** | **0.6036** | **0.8048** | **0.8636** | **0.8801** | **0.3618** | **0.1977** | **0.9117** | **0.1552** | **0.3550** | **0.2436** |
| Purity | Single-Layer | 0.7679 | 0.8576 | 0.7117 | 0.6263 | 0.8946 | 0.8893 | 0.4737 | 0.2371 | 0.7039 | 0.1251 | 0.5379 | 0.3540 |
| | Hierarchical | **0.8556** | **0.9015** | **0.8333** | **0.6288** | **0.9319** | **0.9347** | **0.5758** | **0.2687** | **0.8150** | **0.1348** | **0.6316** | **0.3669** |
| F-score | Single-Layer | 0.6370 | 0.7202 | 0.5484 | 0.4966 | 0.8200 | 0.8343 | 0.3810 | 0.1339 | 0.4052 | 0.0546 | 0.3894 | 0.1078 |
| | Hierarchical | **0.7549** | **0.8632** | **0.7073** | **0.5031** | **0.8714** | **0.8793** | **0.4217** | **0.1393** | **0.7518** | **0.0634** | **0.4743** | **0.1234** |

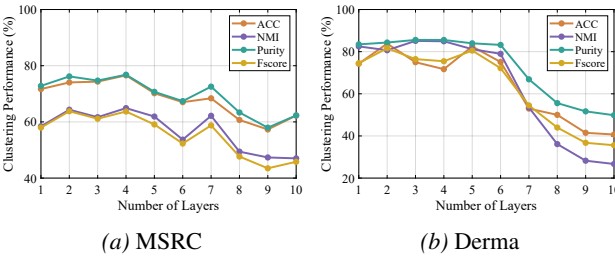

*(a) MSRC*      *(b) Derma*

*Figure 3.* Clustering performance with different numbers of hierarchical layers on MSRC and Derma datasets.

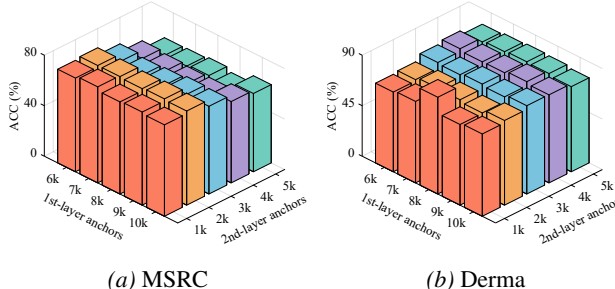

*(a) MSRC*      *(b) Derma*

*Figure 4.* Clustering performance of different anchor combinations on MSRC and Derma datasets.

**Impact of Hierarchy Depth.** Figure 3 evaluates the effect of varying the number of hierarchical layers on clustering performance. The anchor numbers are configured in a descending manner from $10k$ to $k$, where each additional layer removes the largest anchor set from the hierarchy. Compared with the single-layer structure, introducing multiple layers consistently improves performance. The performance gain becomes saturated when the number of layers reaches four, this also experimentally supports our choice of setting the maximum hierarchy depth to $L = 4$, striking a balance between representation richness and computational efficiency.

**Impact of Anchor Combinations.** Figure 4 quantifies the clustering performance under various anchor number configurations in the two-layer setting. While the number of second-layer anchors exhibits a relatively greater influence on performance compared to the first layer, the overall variation in accuracy across different anchor combinations remains small. Moreover, the performance trends are more stable across datasets, indicating that the proposed hierarchical design can maintain robustness against moderate changes in anchor numbers. This further suggests that the learned hierarchical anchor representations inherently possess strong adaptability, enabling the model to preserve discriminative clustering structures without requiring precise manual tuning of anchor numbers at each layer.

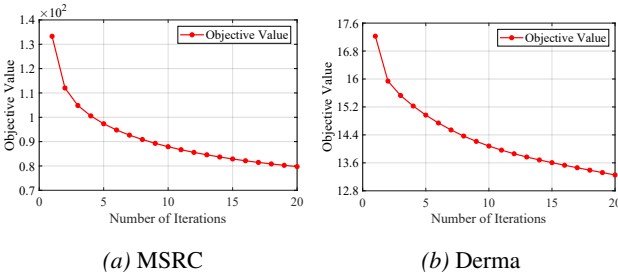

*(a) MSRC*      *(b) Derma*

*Figure 5.* The convergence analysis of our proposed method on MSRC and Derma datasets.

### 4.6. Convergence

Our proposed algorithm is theoretically guaranteed to converge to a local minimum. To empirically validate this property, we monitor the evolution of the objective function value over iterations on two datasets. As illustrated in Figure 5, the objective values consistently exhibit a monotonically decreasing trend and rapidly converge within a small number of iterations, demonstrating the efficiency and stability of our optimization procedure in practice.

## 5. Conclusion

This work challenges single-layer anchor learning by proposing HAG-MVC, a framework that treats multi-view clustering as a hierarchical abstraction process. Inspired by human cognition, it employs a multi-level co-evolution

mechanism to distill semantic prototypes from local details. Unlike "black-box" deep methods, it maintains anchors in the original feature space for transparency and inspectability. Extensive experiments confirm its superior performance and linear scalability. While adaptive depth remains an open challenge, this study establishes a robust foundation for large-scale hierarchical representation.

## Acknowledgements

This work was supported in part by the Natural Science Foundation of Hunan Province, China, under Grant 2026JJ20079, and the National Natural Science Foundation of China (NSFC) under Grants 62376279 and 62306324.

## Impact Statement

This paper presents work whose goal is to advance the field of Machine Learning. There are many potential societal consequences of our work, none of which we feel must be specifically highlighted here.

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

# A. Theoretical Proof for Convergence

In this section, we provide a theoretical proof regarding the convergence of the proposed optimization algorithm for HAG-MVC. Since the objective function $\mathcal{J}$ couples the variables $\mathbf{U}^{(l)}$ and $\mathbf{V}_t^{(l)}$, the optimization problem is non-convex with respect to all variables simultaneously. However, the subproblem with respect to each individual variable is convex when the others are fixed. Therefore, we adopt an alternate optimization strategy that updates each variable while fixing the others. The convergence of the proposed algorithm is proved as follows.

First, we analyze the boundedness of the objective function. The objective function $\mathcal{J}$ is composed of non-negative squared Frobenius norm terms and non-negative regularization terms. Additionally, the constraint $\mathbf{U}^{(l)} \geq 0$ ensures the non-negativity of the graph variables. Consequently, for any iteration, the inequality $\mathcal{J} \geq 0$ always holds, indicating that the objective function is bounded from below.

Second, we analyze the monotonicity of the objective value. Let $\tau$ denote the current iteration index. The optimization process proceeds in two alternating steps:

Update $\mathbf{U}^{(l)}$: When the anchor matrices $\{\mathbf{V}_t^{(l)}\}$ are fixed, the optimization w.r.t. $\mathbf{U}^{(l)}$ is a convex quadratic programming (QP) problem with linear constraints. Our algorithm solves this convex subproblem to obtain the global optimum $\mathbf{U}^{(l)(\tau+1)}$. Therefore, the objective value is guaranteed to be non-increasing:

$$\mathcal{J}(\mathbf{U}^{(l)(\tau+1)}, \mathbf{V}_t^{(l)(\tau)}) \leq \mathcal{J}(\mathbf{U}^{(l)(\tau)}, \mathbf{V}_t^{(l)(\tau)}). \tag{11}$$

Update $\mathbf{V}_t^{(l)}$: When $\{\mathbf{U}^{(l)}\}$ are fixed, the optimization with respect to each $\mathbf{V}_t^{(l)}$ reduces to an unconstrained least-squares problem, which is strictly convex regarding the target variable. The closed-form solution derived by setting the derivative to zero corresponds to the global minimum $\mathbf{V}_t^{(l)(\tau+1)}$. Thus, the following inequality holds:

$$\mathcal{J}(\mathbf{U}^{(l)(\tau+1)}, \mathbf{V}_t^{(l)(\tau+1)}) \leq \mathcal{J}(\mathbf{U}^{(l)(\tau+1)}, \mathbf{V}_t^{(l)(\tau)}). \tag{12}$$

By combining these two steps, each complete iteration ensures that the objective function value is monotonically non-increasing, i.e., $\mathcal{J}^{(\tau+1)} \leq \mathcal{J}^{(\tau)}$. In summary, the sequence of objective function values $\{\mathcal{J}^{(\tau)}\}$ is monotonically non-increasing and bounded from below by 0. According to the Monotone Convergence Theorem, the sequence must converge. Therefore, the proposed optimization algorithm is theoretically guaranteed to converge to a stationary point of the objective function.

# B. Extended Related Work

In this supplement, we provide an extended discussion of the literature to further contextualize our proposed HAG-MVC. The main notations adopted in this study are summarized in Table 5.

## B.1. Anchor-based Multi-View Clustering

To address the scalability issue of multi-view subspace clustering on large-scale datasets, anchor-based multi-view clustering has been proposed, which leverages $c \ll n$ anchors to construct an $n \times c$ bipartite graph for clustering, and reduces the time complexity to $\mathcal{O}(nc)$ (Liu et al., 2025). Traditional anchor-based multi-view clustering methods generally follow a two-step process: constructing individual anchor graphs for each view and integrating them through anchor graph fusion method. A large number of methods have been proposed to explore diverse strategies for anchor graph fusion in multi-view clustering. For example, Kang et al. (Kang et al., 2020) directly assigned the same weight to all views, Zhou et al. (Zhou et al., 2025) employed an auto-weighted allocation strategy to adaptively learn appropriate weight factors for each view, and Li et al. (Li et al., 2020) presented a parameter-free anchor graph fusion method.

In contrast to the two-step fusion strategy, a number of recent studies have attempted to bypass the explicit construction of anchor graphs for individual views. Instead, they directly optimize a unified objective to obtain the fused anchor graph, and reduce intermediate computation and potential inconsistencies across views. The classical anchor-based multi-view clustering method that directly learns a fused anchor graph from multiple views can be expressed as follows:

$$\min_{\mathbf{U}} \sum_{t=1}^{M} \|\mathbf{X}_t - \mathbf{V}_t \mathbf{U}\|_F^2 + \mu \|\mathbf{U}\|_F^2$$

$$\text{s.t. } \mathbf{U}^\top \mathbf{1}_m = \mathbf{1}_n, \mathbf{U} \geq 0, \tag{13}$$

*Table 5.* The Main Notations

| Notations | Description |
|---|---|
| $n$ | Number of instances |
| $k$ | Number of clusters |
| $c_l$ | Number of anchors at the $l$-th layer |
| $M$ | Number of views |
| $\mu_l$ | Regularization parameter at the $l$-th layer |
| $\mathbf{X}_t \in \mathbb{R}^{d_t \times n}$ | Input data matrix in the $t$-th view |
| $\mathbf{V}_t^{(l)} \in \mathbb{R}^{d_t \times c_l}$ | Anchor matrix in the $t$-th view at the $l$-th layer |
| $\mathbf{U}^{(1)} \in \mathbb{R}^{c_1 \times n}$ | Sample-to-anchor bipartite graph at the first layer |
| $\mathbf{U}^{(l)} \in \mathbb{R}^{c_l \times c_{l-1}}$ | Cross-layer anchor graph from $(l{-}1)$-th to $l$-th layer |
| $\mathbf{U} \in \mathbb{R}^{n \times c_L}$ | Final consensus bipartite graph |

where $\mathbf{V}_t$ denotes the anchor matrix in the $t$-th view, $\mathbf{U}$ denotes the consensus anchor graph and the constraint $\mathbf{U}^\top \mathbf{1}_m = \mathbf{1}_n$ guarantees that the total similarity between each sample and all anchors is normalized to 1.

Note that $\mathbf{V}_t$ is fixed before the optimization process, so the effectiveness of clustering algorithms is highly dependent on the quality of anchor points, making the selection of representative anchors a critical research focus. Early approaches typically employed random sampling from the original dataset to generate anchors, which often suffers from instability issues. Therefore, heuristic strategies have been developed in several studies such as $k$-means based methods and feature score based schemes (Yang et al., 2024; 2022). For example, Li et al. (Li et al., 2020) provided a direct alternate sampling method to determine the anchors based on scores. In contrast to these direct anchor selection methods, recent research has shifted toward adaptive anchor learning, which simultaneously optimizes latent anchor points and subspace representations within a unified framework. For example, Sun et al. (Sun et al., 2021) learned common anchors and the corresponding anchor graph through a projection-based approach. Wen et al. (Wen et al., 2023) proposed learning the view-specific anchor and the consistent anchor graph by leveraging local and global structural information.

The final clustering indicator matrix can be calculated from the SVD of $\mathbf{U}$ (Kang et al., 2020; Xie et al., 2019). Consequently, the computational and space expenditure is reduced from $\mathcal{O}(n^3)$ to $\mathcal{O}(Mnc)$, where $n$, $c$, and $M$ denote the number of samples, anchors, and views, respectively.

## B.2. Hierarchical Clustering

Hierarchical clustering has been employed to model multi-level structural relationships in data. Classical methods, such as agglomerative and divisive clustering, recursively merge or split samples to form tree-like structures that capture nested dependencies (Wang et al., 2021; Agarwal et al., 2022; Monath et al., 2021; Long & van Noord, 2023). While effective for low-dimensional or small-scale tasks, these methods suffer from poor scalability and sensitivity to local similarity measures, limiting their performance in high-dimensional, multi-view, and large-scale scenarios (Han et al., 2022; Dhulipala et al., 2024).

In the field of multi-view clustering, several studies adopt deep matrix factorization frameworks, which learn layered low-rank subspace representations to uncover shared structures across views. Although these methods do not explicitly construct hierarchical graphs, their layer-wise factorization can be regarded as an implicit form of hierarchical modeling, progressively extracting semantic abstractions from low-level features to high-level representations (Deng et al., 2023; Wang et al., 2023; Feng et al., 2024). Such approaches have shown notable improvements in clustering performance.

However, deep factorization models often lack structural interpretability, as they do not explicitly model hierarchical dependencies or support inter-layer structural propagation. Their optimization is typically complex and sensitive to hyperparameters, limiting generalization in practical applications. To address these issues, some studies have explored anchor-based hierarchical clustering frameworks that explicitly organize anchors into multiple layers and construct bipartite graphs across layers to facilitate structural abstraction and semantic consistency. For example, Wang et al. (Wang et al., 2017) first introduced the hierarchical anchor graph into semi-supervised learning, and Li et al. (Yang et al., 2020) further extended this concept by constructing a hierarchical anchor graph within an unsupervised learning framework. While these

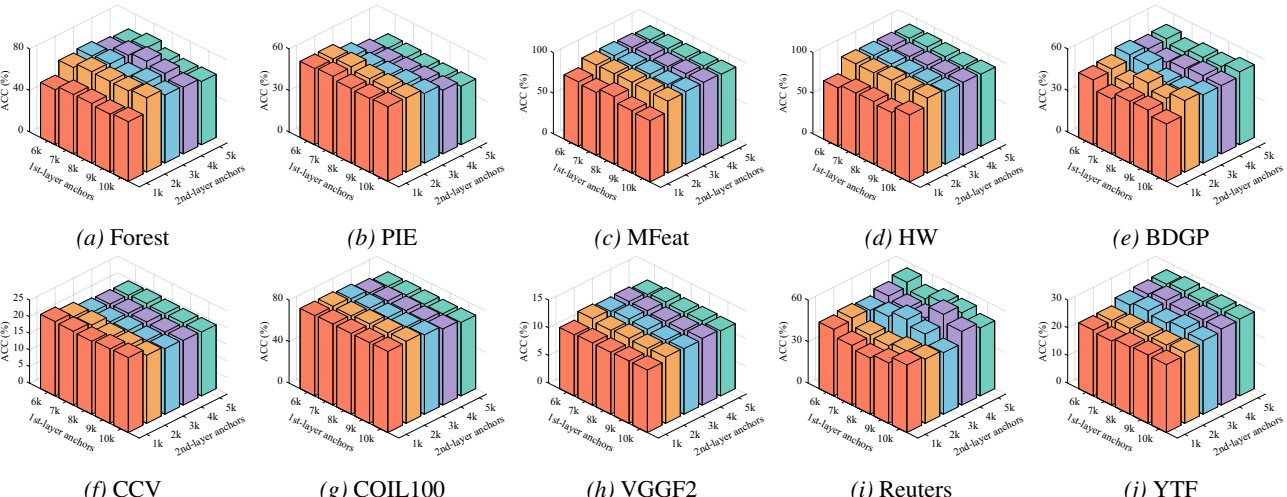

*Figure 6.* Clustering performance of different anchor combinations on the remaining ten datasets.

methods employ hierarchical structures, they are primarily designed for single-view data. Recently, hierarchical modeling has been incorporated into anchor-based multi-view clustering methods. For example, Zhou et al. (Zhou et al., 2025) proposed generating anchor layers using bisecting $k$-means and constructing hierarchical bipartite graphs. Nevertheless, these methods rely on $k$-means initialization to generate anchors, which remain fixed throughout the learning process, potentially preventing the selected anchors from adequately capturing the intrinsic structure of the data or adapting to variations across views.

## C. Detailed Description of Benchmark Datasets

In our experiments, we conduct extensive evaluations on twelve widely-used multi-view benchmark datasets, including MSRC, Dermatology (Derma), ForestTypes (Forest), PIE, MFeat, HandWritten (HW), BDGP, CCV, COIL100, VGGFace2 (VGGF2), Reuters, and YoutubeFace (YTF). These datasets span a wide range of domains such as natural images, handwritten digits, biomedical data, text, and face videos. The sample sizes range from a few hundred to over one hundred thousand, with the number of views varying from 2 to 6 and the number of categories ranging from 4 to 100.

Specifically, MSRC is an image dataset consisting of 210 samples from 7 categories, represented by five visual views including CM, HOG, GIST, LBP and CENT features. The Dermatology dataset contains diagnostic information for six types of skin diseases, where the two views respectively describe the clinical diagnostic information and the histopathological information. ForestTypes consists of satellite images from different forest types represented by three low-dimensional views. PIE is constructed from the CMU Multi-PIE face database using 32×32 facial images from sessions C07, C09, and C29 to form a three-view dataset. HandWritten contains 2000 samples for digit recognition across 10 classes, represented by six different feature views. MFeat includes 2000 images of handwritten digits ranging from 0 to 9. BDGP includes Drosophila embryo images captured from lateral, dorsal, and ventral views. CCV is a video collection from YouTubeFace that covers 20 high-level semantic categories. COIL100, VGGFace2, Reuters, and YoutubeFace are high-dimensional, large-scale datasets.

## D. Other Experimental Results

### D.1. Running Time Comparison

Despite the clear advantages in clustering performance, the proposed HAG-MVC also maintains competitive efficiency compared to existing state-of-the-art methods. As shown in Figure 9, we conduct a systematic comparison of the running time between HAG-MVC and several baseline methods across all benchmark datasets, where the $y$-axis is shown on a logarithmic scale and an empty bar indicates out-of-memory (OOM). Although adopting a multi-layer anchor structure, HAG-MVC maintains efficiency by using far fewer anchors than the total number of samples at each layer. And it is worth noting that all compared algorithms, including our method, operate with linear time complexity in the number of samples, ensuring scalability to large-scale datasets.

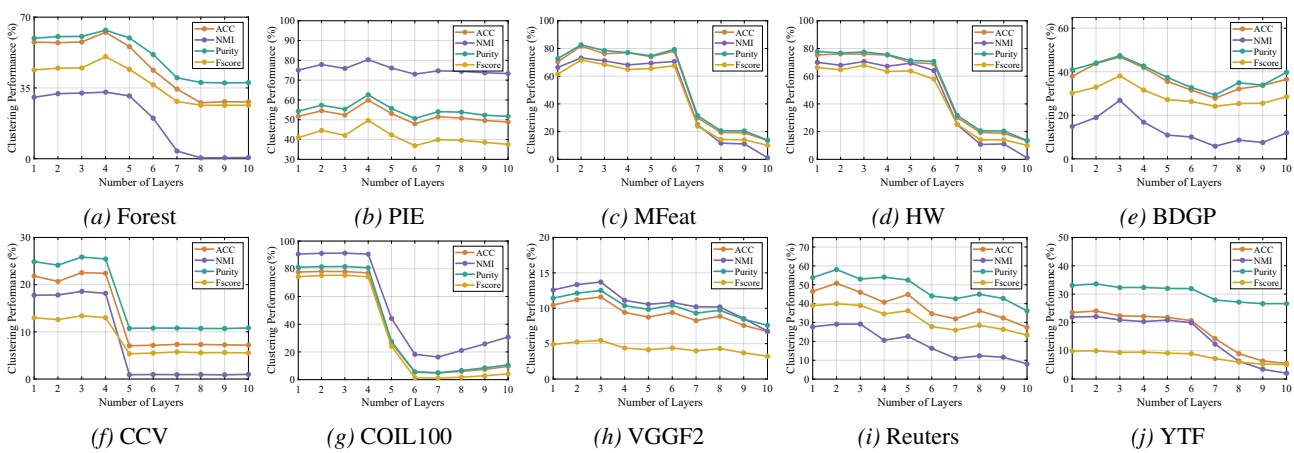

*Figure 7.* Clustering performance with different numbers of hierarchical layers on the remaining ten datasets.

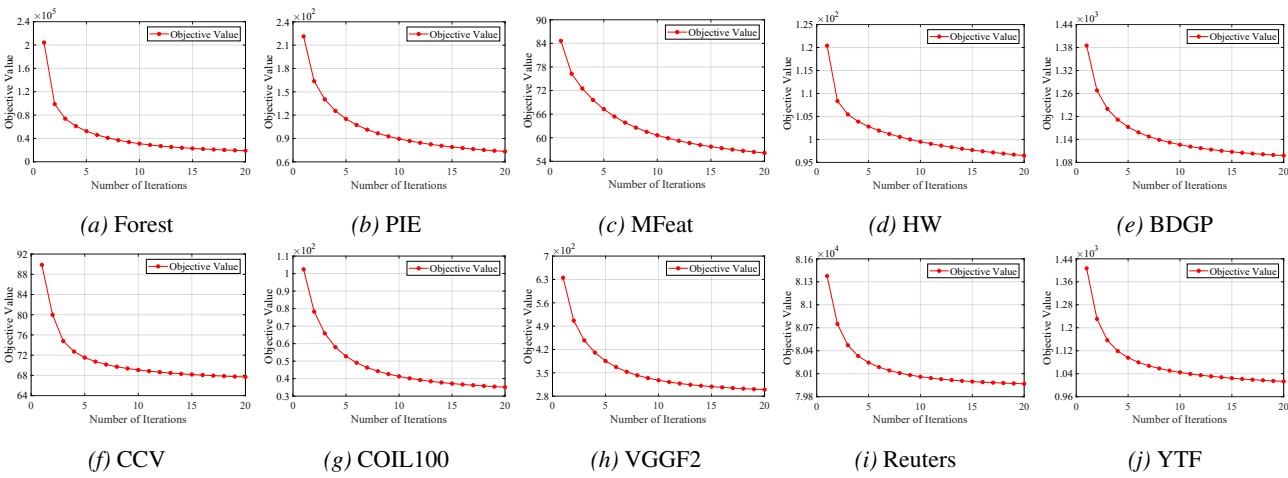

*Figure 8.* The convergence analysis of our proposed method on the remaining ten benchmark datasets.

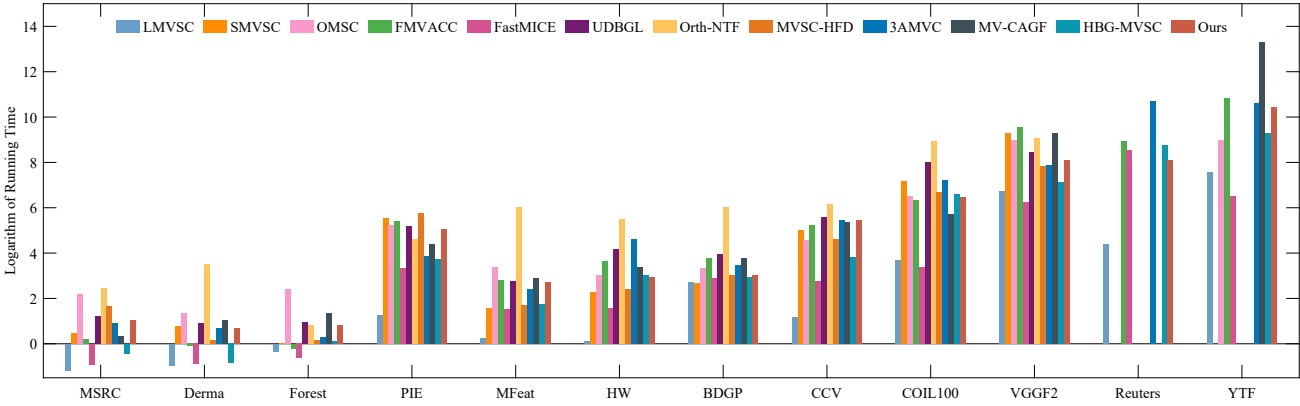

*Figure 9.* Running time comparison of different methods.

## D.2. Detailed Experimental Results of Sec. 4.5

Due to space limitations in the main text, we presented the parameter sensitivity analysis on a subset of representative datasets. In this section, we provide a comprehensive evaluation on the remaining ten datasets to further verify the robustness and generalization capability of the proposed HAG-MVC framework.

**Robustness to Anchor Number Configurations.**   We first present the comprehensive sensitivity analysis regarding the number of anchors in the two layers on the remaining ten datasets. As shown in Figure 6, the clustering performance exhibits a high degree of stability over a broad range of anchor combinations. Notably, while the performance surface is generally flat, the sensitivity to the number of anchors in the second layer is slightly more pronounced than that of the first layer in datasets with complex semantic structures. This observation aligns with our hierarchical design motivation: the deeper layers abstract higher-level semantic concepts that are more critical for the final partitioning. Overall, the broad optimal parameter range demonstrates the robustness of HAG-MVC, relieving the need for precise hyperparameter tuning in practical applications.

**Impact of Hierarchy Depth.**   Secondly, we investigate the impact of hierarchy depth ($L$) on representation learning quality. The performance trends across all datasets consistently show that stacking hierarchical layers yields significant improvements over the single-layer baseline ($L = 1$). This enhancement stems from the progressive semantic abstraction capability inherent in the hierarchical design. By projecting data onto a sequence of increasingly compact anchor sets, the model effectively filters out local variations and low-level noise, thereby distilling more discriminative and globally consistent structural information. Furthermore, we observe that the performance gains typically saturate at $L = 4$. Increasing the depth beyond this point ($L \geq 5$) offers negligible improvements and may risk introducing over-smoothing effects. Consequently, a four-layer hierarchy strikes an optimal balance between representation richness and computational efficiency, serving as the recommended configuration for our framework.

### D.3. Detailed Experimental Results of Sec. 4.6

To empirically verify the numerical stability and optimization efficiency of HAG-MVC, we record the objective function values across the iterations on the remaining ten datasets. As illustrated in Figure 8, the objective function value exhibits a sharp and monotonic decrease during the initial phase across all benchmarks, including large-scale ones like Reuters and YoutubeFace. Following this rapid descent, the curves quickly stabilize and reach a stationary point within 15 to 20 iterations without significant oscillations. This fast convergence speed is attributed to our efficient alternating minimization strategy, where the optimization problem is convex with respect to each variable block while fixing the others, thereby theoretically guaranteeing convergence to a local optimum and ensuring low computational overhead in practice.

