# OpenReview forum: "Hierarchical Anchor Graph Learning for Multi-View Clustering"
_ICML.cc/2026/Conference — ICML 2026 regular_

### Official Review · Reviewer_iR4n · 2026-03-03

**Soundness:** 3
**Presentation:** 3
**Significance:** 3
**Originality:** 3
**Overall Recommendation:** 5
**Confidence:** 4

**Summary:**

This paper studies anchor-based multi-view clustering under a hierarchical formulation. It argues that existing methods mostly rely on a single fixed anchor layer, which limits their ability to capture multi-level structure in complex data. To address this, the paper proposes HAG-MVC, which organizes anchors into multiple layers and jointly learns the anchor representations and cross-layer graph structure. The method uses lower layers to model sample-to-anchor relations and higher layers to progressively form more compact semantic representations, while keeping anchors in the original feature space. The paper also presents an alternating optimization algorithm, along with convergence and complexity analysis. Experiments show strong performance against recent baselines.

**Compliance With Llm Reviewing Policy:**

Affirmed.

**Final Justification:**

The rebuttal addressed my concerns.

**Key Questions For Authors:**

1. The paper shows that the gains from increasing hierarchy depth become quite limited after around four layers. It would be helpful if the authors could explain what they see as the main limiting factor here, such as over smoothing, redundant abstraction, or simply diminishing returns from adding more layers.
2. The experimental comparison is already strong within the anchor based MVC setting, but the paper would be easier to position if it included a clearer discussion of how this method relates to representative deep multi view clustering approaches, especially in terms of scalability, interpretability, and likely performance tradeoffs.
3. In the current framework, both the hierarchy depth and the anchor numbers are selected from predefined candidate sets. A brief discussion of whether this part could become more adaptive in future work, for example through a more data driven hierarchy design, would make the paper more complete.
4. The runtime results suggest that the method remains efficient even when multiple layers are used. Some added intuition about which part of the computation tends to become the main cost as the hierarchy gets deeper would make this section easier to interpret.

**Limitations:**

Yes

**Strengths And Weaknesses:**

Strengths: The paper is built around a clear and meaningful observation: most existing anchor-based multi-view clustering methods rely on a fixed single-layer anchor structure, which can be restrictive when the data has multi-level semantics. The proposed hierarchical design addresses this limitation in a direct and coherent way, and the full pipeline is consistent. The supporting analysis is also reasonably complete, including convergence discussion, complexity analysis, and an additional theoretical explanation for why the hierarchical graph can capture higher-order relationships. The method is evaluated on twelve benchmark datasets against a strong set of recent anchor-based baselines, and the reported gains are consistently strong. Overall, the work presents a meaningful and well-executed extension of anchor-based MVC that improves modeling flexibility while remaining scalable and interpretable.

Weaknesses: The optimization section is somewhat dense, and the notation could be streamlined to make the derivations easier to follow. The experiments are convincing within the anchor-based MVC setting, but the paper would benefit from a clearer discussion of how it relates to broader deep or hierarchical clustering approaches. It would also help to say a bit more about when the added hierarchy is most beneficial in practice. These issues do not affect the core technical contribution, but addressing them would improve clarity and positioning.

---

> ### Author Rebuttal · Authors · 2026-03-31
>
> We deeply appreciate your positive feedback and valuable suggestions to refine our work.
>
> **W1 (Notation):** To improve readability, we will streamline the mathematical notation and move the heavy intermediate derivations to the Appendix, keeping the main text strictly focused on the core optimization logic.
>
> **W2 & Q2 (Relation to Deep MVC):** To clarify our research positioning, we will expand the Related Work to systematically compare HAG-MVC with representative deep multi-view clustering approaches across three key dimensions:
>  **Interpretability:** Deep models typically rely on uninterpretable "black-box" neural layers in latent spaces. In contrast, HAG-MVC constructs transparent, explicit bipartite graphs directly within the original continuous feature space, allowing for clear error diagnosis.
>  **Scalability:** Deep methods are computationally expensive and heavily dependent on GPU resources. HAG-MVC remains exceptionally CPU-friendly with a strict linear complexity of $\mathcal{O}(n)$, making it highly practical for large-scale datasets.
>  **Performance Tradeoffs:** While deep models excel at capturing highly non-linear mappings, they often suffer from hyperparameter sensitivity and unstable training. HAG-MVC trades complex non-linearity for mathematical transparency, stable convergence, and zero GPU requirement, while still achieving highly competitive clustering accuracy.
>
> **W3 (When Hierarchy is Most Beneficial):** In the revised manuscript, we will explicitly clarify that the added hierarchy is most beneficial in two main practical scenarios. First, for complex data exhibiting high intra-class variance or multi-granularity semantics, such as visual images with rich sub-topics, flat anchors often fail to capture structural nuances, whereas our hierarchy excels by progressively aggregating concepts from fine to coarse. Second, in high-noise environments, our spectral analysis demonstrates that the layer-by-layer graph construction acts as a structural low-pass filter, effectively smoothing out view-specific high-frequency noise and local outliers to robustly uncover the true global consensus manifold.
>
> **Q1 (Limiting Factors):** The performance plateau after about four layers is indeed caused by a combination of over-smoothing and redundant abstraction. As proven by the new spectral analysis detailed in our response to Reviewer TmHk (W1), each bipartite graph layer acts as a low-pass filter. While initial layers successfully remove noise, excessive layers repeatedly apply this smoothing, eventually stripping away the essential high-frequency structural signals needed to distinguish clusters and forcing disparate sub-clusters to merge. We will add this theoretical clarification to the revised manuscript.
>
> **Q3 (Adaptivity):** The current requirement to manually set the hierarchy depth and per-layer anchor sizes indeed introduces a model-selection burden. Making these structural choices adaptive is a crucial step for practical application, and we highly appreciate this forward-looking suggestion. In the revised manuscript, we will add a dedicated discussion on how to evolve HAG-MVC into a fully adaptive framework. Specifically, by introducing sparsity-inducing norms (e.g., $\ell_{2,1}$-norm regularization) on the bipartite graph weight matrices $\mathbf{U}^{(l)}$, the model could automatically zero out irrelevant connections during the alternating optimization. This mechanism would intrinsically prune redundant anchors and seamlessly bypass unnecessary layers, allowing the network to dynamically determine its own optimal width and depth. We will highlight this principled adaptive mechanism as a highly promising direction for future work to entirely eliminate manual hyperparameter dependence.
>
> **Q4 (Computational Cost):** As the hierarchy deepens, the primary additional cost stems from the iterative optimization of anchor graphs between adjacent layers and the anchors themselves across different layers. Since the number of anchors in each layer is significantly smaller than the total number of samples $n$, this computational overhead remains exceptionally lightweight.

---

> > ### Author Rebuttal · Reviewer_iR4n · 2026-04-02
> >
> > The rebuttal addressed all my concerns.

---

> > > ### Author Response · Authors · 2026-04-06
> > >
> > > We sincerely appreciate your careful re-evaluation and supportive comments. Thank you for recognizing our efforts to improve the manuscript.

---

### Official Review · Reviewer_8hXk · 2026-03-08

**Soundness:** 4
**Presentation:** 3
**Significance:** 3
**Originality:** 3
**Overall Recommendation:** 5
**Confidence:** 4

**Summary:**

The paper addresses multi-view clustering with anchor graphs and focuses on a limitation of existing methods: most of them use a single static anchor layer. The proposed HAG-MVC replaces this flat design with a hierarchical anchor graph, where sample-to-anchor and anchor-to-anchor relations are learned across multiple layers. The framework jointly updates anchor representations and graph assignments, while keeping anchors in the original feature space. The results suggest that the hierarchical formulation improves clustering performance over recent anchor-based methods.

**Compliance With Llm Reviewing Policy:**

Affirmed.

**Final Justification:**

I have read the authors' rebuttal response, and I decide to remain my score (accept) unchanged.

**Key Questions For Authors:**

(1) The current framework still requires the user to set the hierarchy depth and the anchor size at each layer. A brief discussion of whether these structural choices could be made more adaptive would make the method more practical.
(2) Could the authors clarify how they would position HAG-MVC relative to hierarchical ideas used in representative deep multi-view clustering methods?
(3) The method remains efficient in the reported experiments, but the hierarchical design also adds model-selection effort. It would help to provide a practical default rule, or at least a simple guideline, for choosing the number of layers and anchor scales.

**Limitations:**

Yes

**Strengths And Weaknesses:**

Pros
- Addresses a critical issue in scalable multi-view clustering and clearly identifies the key limitation of existing anchor-based approaches (reliance on a single static anchor layer).
- Proposes a reasonable hierarchical anchor graph extension with a consistent model, optimization, and clustering pipeline.
- Delivers a meaningful hierarchical extension that preserves scalability and interpretability for anchor-based multi-view clustering.

Cons:
- Hierarchical structure adds extra model-selection burden (users must manually set the number of layers and anchor size per layer).
- No principled method to reduce dependence on manual hyperparameter settings, even with sensitivity analysis.
- Related work is overly focused on anchor-based multi-view clustering, with limited discussion of hierarchical concepts in deep multi-view clustering.
- Lack of broad conceptual comparison weakens the clarity of the paper’s research positioning.

---

> ### Author Rebuttal · Authors · 2026-03-31
>
> We sincerely thank you for the positive evaluation and highly constructive feedback on our work.
>
> **W1 & Q1 (Structural Adaptivity):** The current requirement to manually set the hierarchy depth and per-layer anchor sizes indeed introduces a model-selection burden. Making these structural choices adaptive is a crucial step for practical application, and we highly appreciate this forward-looking suggestion. In the revised manuscript, we will add a dedicated discussion on how to evolve HAG-MVC into a fully adaptive framework. Specifically, by introducing sparsity-inducing norms (e.g., $\ell_{2,1}$-norm regularization) on the bipartite graph weight matrices $\mathbf{U}^{(l)}$, the model could automatically zero out irrelevant connections during the alternating optimization. This mechanism would intrinsically prune redundant anchors and seamlessly bypass unnecessary layers, allowing the network to dynamically determine its own optimal width and depth. We will highlight this principled adaptive mechanism as a highly promising direction for future work to entirely eliminate manual hyperparameter dependence.
>
> **W2 & Q3 (Practical Guidelines for Model Selection):** To address the model-selection effort and provide a principled method for reducing manual hyperparameter dependence, we will explicitly include the following practical default rules and search space reduction strategies in the revised manuscript:
>
> 1. **For the hierarchy depth ($L$):** Based on our newly added spectral analysis, setting $L \in [3,4]$ serves as a robust default rule. This depth optimally filters out high-frequency multi-view noise without triggering the over-smoothing effect, largely eliminating the need for an exhaustive depth search.
> 2. **For the anchor scales:** Once the depth is fixed, the actual hyperparameter search space for anchor numbers is mathematically much smaller than it appears. In practice, we only traverse a small candidate set from $k$ to $5k$ (e.g., $\{k, 2k, 3k, 4k, 5k\}$). Crucially, the hierarchical design dictates that the number of anchors must **strictly decrease layer by layer** ($c_1 > c_2 > \dots > c_L$). Taking a 3-layer network ($L=3$) as an example, selecting 3 strictly decreasing values from 5 candidates yields only $\binom{5}{3} = 10$ valid combinations. This structural constraint drastically shrinks the search space, providing a simple yet highly effective guideline that makes the practical tuning process computationally trivial.
>
> **W3 & Q2 (Conceptual Positioning Relative to Deep Hierarchical MVC):** We appreciate this insightful suggestion to broaden our conceptual comparison. In the revised manuscript, we will expand the Related Work section to explicitly discuss hierarchical concepts in deep multi-view clustering and clarify our research positioning.
>
> Conceptually, representative deep multi-view clustering methods build hierarchies by utilizing nested non-linear neural layers such as MLPs and GCNs to extract abstract feature representations. However, this hierarchical abstraction occurs within an uninterpretable latent space, inherently suffering from "black-box" opacity and heavy reliance on GPU resources.
>
> In contrast, we position HAG-MVC as a transparent, mathematically interpretable, and computationally lightweight alternative. Our method constructs a hierarchy of structural representations directly within the original continuous feature space. The hierarchy in HAG-MVC operates as an explicit chain of bipartite graphs that progressively groups samples into fine-to-coarse semantic prototypes. This expansion clearly distinguishes our work from deep multi-view clustering, demonstrating that HAG-MVC successfully integrates the powerful representation capacity of hierarchical learning with the transparency and CPU-friendly efficiency of traditional anchor-based methods.
>
> **W4 (Broad Conceptual Comparison):** To address the need for a clearer research positioning, we will add a dedicated subsection and a conceptual comparison table in the revised manuscript to systematically contrast HAG-MVC across three major paradigms:
> 1. **Traditional Anchor-based MVC:** Limited by flat structures and statically fixed anchors.
> 2. **Deep Hierarchical MVC:** Builds hierarchies via nested nonlinear neural networks such as MLPs and GCNs, but suffers from "black-box" opacity, uninterpretable latent spaces, and heavy GPU resource reliance.
> 3. **HAG-MVC (Ours):** Uniquely achieves hierarchical abstraction directly within the original continuous feature space. It provides a transparent, mathematically interpretable multi-level co-evolution mechanism while remaining computationally lightweight and CPU-friendly.

---

> > ### Author Rebuttal · Reviewer_8hXk · 2026-04-03
> >
> > This is a good paper. Thanks for addressing all my concerns.

---

> > > ### Author Response · Authors · 2026-04-06
> > >
> > > Thank you very much for your encouraging feedback and for taking the time to revisit our response. We are glad that our clarifications were helpful and addressed your concerns. We sincerely appreciate your careful review and valuable suggestions.

---

### Official Review · Reviewer_Bd1f · 2026-03-10

**Soundness:** 4
**Presentation:** 3
**Significance:** 3
**Originality:** 3
**Overall Recommendation:** 5
**Confidence:** 4

**Summary:**

The paper proposes HAG-MVC, a hierarchical anchor-graph framework for multi-view clustering that replaces the common single-layer, static anchor set with a multi-level pyramid of anchors. The method jointly refines (i) sample-to-anchor assignments at the first layer and (ii) anchor-to-anchor graphs across higher layers via an alternating optimization procedure, producing a final fused anchor graph for clustering.

**Compliance With Llm Reviewing Policy:**

Affirmed.

**Final Justification:**

The authors’ response has addressed my concerns, and I will raise my score.

**Key Questions For Authors:**

a)The complexity analysis includes terms like “solving n QP subproblems” and cubic-in-anchor terms. Can the authors provide runtime/memory curves and iteration counts to support the practical scaling claim
b)Table 2 reports many N/A baselines due to execution failure. What were the resource limits, and can the authors provide comparison against the runnable scalable baselines?
c)The paper emphasizes interpretability. Do the authors have real data studies demonstrating actionable “error diagnosis” as claimed?

**Limitations:**

Limitations can be found in the Weakness and Questions.

**Strengths And Weaknesses:**

Strength
a)The hierarchical “fine-to-coarse” abstraction idea is intuitive for MVC and aligns well with anchor-graph scalability goals. The paper explicitly couples anchor representations and assignment/transition graphs in a unified objective with alternating updates.
b)The experiment in the paper includes many datasets, multiple metrics, and meaningful ablations, validating the effectiveness of the proposed method.

Weakness
a)The paper should more sharply distinguish itself from existing hierarchical anchor / hierarchical MVC baselines and clarify what is fundamentally new beyond “multi-layer anchors + alternating refinement.
b)While the paper argues near-linear scaling, the optimization involves solving many QP subproblems (column-wise) and includes cubic terms in anchor size; practical scalability depends heavily on solver choices and iteration counts.
c)The manuscript contains clear editorial/structural anomalies, such as a duplicated “Introduction” section numbering (“1. Introduction” immediately followed by “2. Introduction”).

---

> ### Author Rebuttal · Authors · 2026-03-31
>
> We sincerely thank you for the insightful comments and constructive suggestions on our work.
>
> **Wa (Novelty vs. Baselines):** While existing hierarchical multi-view clustering methods like HBG-MVSC construct multi-layer graphs, they typically rely on statically fixed anchors pre-generated layer-by-layer using heuristic algorithms like $k$-means. In those paradigms, the hierarchical structure is rigidly dependent on the quality of the initial sampling, and errors at the bottom layers propagate upward irreversibly. What is fundamentally new in HAG-MVC beyond just "multi-layer anchors + alternating refinement" is the multi-level co-evolution mechanism within the original continuous feature space. We treat the anchor coordinates $\mathbf{V}^{(l)}$ themselves as dynamically learnable variables across all layers, rather than fixed reference points. Our alternating procedure does not merely update the bipartite graph weights $\mathbf{U}^{(l)}$; it physically shifts the anchor prototypes to iteratively align with the emerging cross-view semantic consensus. This dynamic refinement empowers the model to continuously correct suboptimal local densities and adaptively discover true multi-granularity prototypes from fine to coarse. We will explicitly highlight this fundamental distinction in the Introduction and Related Work sections of the revised manuscript.
>
> **Wb & Qa (Practical Scalability & Runtime):** We agree that practical scaling depends heavily on solver efficiency and iteration counts. Regarding the need to solve $n$ QP subproblems, since optimizing the assignment for each sample is strictly independent when anchors are fixed, these subproblems are trivially parallelizable. In our MATLAB implementation, we utilize `parfor` to solve them simultaneously, which drastically reduces the actual wall-clock time. Regarding the cubic term $\mathcal{O}(c^3)$, the number of anchors $c$ is defined as a small multiple of the true number of clusters $k$ (e.g., from $10k$ down to $k$). Because $k$ is inherently small, $c \ll n$ strongly holds true, making the $c^3$ term a negligible constant in practice and ensuring the linear term $\mathcal{O}(n)$ strictly dominates. Our optimization converges rapidly within 15-20 iterations as shown in Appendix Figure 8. To firmly support our scalability claim, the actual running times of HAG-MVC are provided below. Furthermore, Appendix Figure 9 provides a comprehensive runtime comparison across all datasets, demonstrating that HAG-MVC maintains highly competitive execution times alongside other linear-time baselines. Even for the YTF dataset with over 100,000 samples, our algorithm finishes efficiently.
>
> | Dataset | Time (s) | Dataset | Time (s) |
> | :--- | :--- | :--- | :--- |
> | MSRC | 2.06 | BDGP | 8.23 |
> | Derma | 1.60 | CCV | 43.01 |
> | Forest | 1.75 | COIL100 | 88.45 |
> | PIE | 32.95 | VGGF2 | 273.88 |
> | MFeat | 6.57 | Reuters | 272.76 |
> | HW | 7.68 | YTF | 1401.61 |
>
> **Wc (Editorial Anomalies):** We apologize for the typographical error regarding the duplicated "Introduction" section numbering. We have carefully proofread the manuscript and corrected this structural anomaly, along with other minor typos, for the revised version.
>
> **Qb (Resource Limits & N/A Baselines):** The "N/A" entries in Table 2 indicate Out-Of-Memory (OOM) execution failures. Our experiments were conducted on a standard machine equipped with 32GB of RAM. Many traditional or unscalable baselines require constructing full $n \times n$ affinity matrices or dense tensor operations, leading to $\mathcal{O}(n^2)$ or $\mathcal{O}(n^3)$ memory complexity. This instantly exceeds the 32GB memory limit on large-scale datasets such as Reuters which has a total feature dimension exceeding 100,000 and YTF which contains over 100,000 samples. To ensure a more comprehensive and fair evaluation, we will replace these OOM-prone methods with recent, runnable, and scalable multi-view clustering baselines in the revised manuscript.
>
> **Qc (Error Diagnosis):** While Section 4.1 demonstrated this transparency on synthetic data, we agree that a real-world case study will significantly strengthen our claim. In the revised manuscript, specifically in the Appendix, we will add an actionable error diagnosis study using the HandWritten dataset. Specifically, we will trace the bipartite connectivity paths for misclassified samples. By explicitly tracking the assignment path $x_i \to \mathbf{v}_a^{(1)} \to \dots \to \mathbf{v}_k^{(L)}$, we can pinpoint exactly where the clustering error occurred. For instance, we can diagnose whether a digit 7 misclassified as a 1 originated from ambiguous local feature extraction at the bottom layer, i.e., assigning to the wrong low-level prototype, or from an incorrect semantic aggregation at the top layer. This real-data visualization will concretely validate the actionable interpretability of HAG-MVC.

---

> > ### Author Rebuttal · Reviewer_Bd1f · 2026-04-01
> >
> > The authors’ response has addressed my concerns, and I will raise my score.

---

> > > ### Author Response · Authors · 2026-04-06
> > >
> > > Thank you very much for your thoughtful follow-up and for recognizing our efforts in revising the manuscript and addressing your concerns. We sincerely appreciate your time, careful evaluation, and encouraging feedback. Your comments have been very helpful in improving the quality and clarity of our work.

---

### Official Review · Reviewer_TmHk · 2026-03-12

**Soundness:** 3
**Presentation:** 4
**Significance:** 3
**Originality:** 3
**Overall Recommendation:** 4
**Confidence:** 3

**Summary:**

This paper proposes a hierarchical anchor graph learning framework for multi-view clustering, termed HAG-MVC. The method aims to address limitations of traditional anchor-based multi-view clustering approaches, which typically rely on a static single-layer anchor set and may fail to capture hierarchical semantic structures in complex multi-view data. Extensive experiments on twelve widely used multi-view datasets demonstrate that the proposed method achieves improved clustering performance compared with several state-of-the-art anchor-based multi-view clustering approaches. The method also maintains linear computational complexity with respect to the number of samples, making it suitable for large-scale datasets.

**Compliance With Llm Reviewing Policy:**

Affirmed.

**Final Justification:**

The author completely addressed my concerns.

**Key Questions For Authors:**

1.The experiments suggest that performance improvements saturate when the hierarchy depth reaches a certain level. Could the authors provide more theoretical or empirical guidance on how to determine the optimal number of hierarchical layers?

2.The method still relies on initialization strategies such as k-means for anchor initialization. How sensitive is the performance to different initialization schemes?

3.Could the hierarchical anchor framework be extended to other graph-based learning tasks such as classification or representation learning?

**Limitations:**

The proposed method relies on hierarchical anchor structures whose effectiveness may depend on appropriate choices of anchor numbers and hierarchy depth. Although the authors demonstrate that the method is relatively robust to these parameters, the hierarchical design may still introduce additional tuning complexity in practice. Furthermore, the current evaluation focuses primarily on anchor-based clustering baselines, and comparisons with more recent deep multi-view representation learning methods would provide a clearer understanding of the method’s relative advantages. Finally, while the model emphasizes interpretability, the theoretical understanding of why hierarchical anchors improve clustering performance could be further strengthened.

**Strengths And Weaknesses:**

Strengths

1.The paper clearly identifies the limitations of existing anchor-based multi-view clustering methods, particularly their reliance on static single-layer anchors that fail to capture hierarchical semantic structures.

2.The proposed hierarchical anchor graph design enables the model to represent multi-scale structures through progressive abstraction, improving clustering representation capability.

3.The multi-level co-evolution mechanism allows anchors and anchor graphs to be iteratively refined together, which improves anchor stability and representation quality compared to fixed-anchor strategies.

4.Experiments are conducted on twelve benchmark datasets spanning multiple domains (e.g., images, text, biomedical data), demonstrating consistent performance improvements over several strong baselines.


5.The framework maintains linear computational complexity and keeps anchors in the original feature space, which enhances interpretability compared to black-box deep clustering methods.

 Weaknesses

1.While the paper provides some theoretical justification related to high-order proximity, the theoretical insights remain relatively limited and could be strengthened.

2.Although the method is theoretically linear in sample size, increasing the number of hierarchical layers may still introduce additional optimization complexity.

3.The experimental comparisons mainly focus on anchor-based clustering methods. Including more recent deep multi-view clustering or representation learning baselines would provide a more comprehensive evaluation.

---

> ### Author Rebuttal · Authors · 2026-03-31
>
> We sincerely thank you for the highly encouraging evaluation.
>
> **W1&Q1 (Theoretical Analysis & Optimal Depth):** In the revised manuscript, we will introduce spectral graph theory to formally prove that our hierarchical architecture acts as a structural low-pass filter. The multi-layer bipartite graph chain effectively approximates raising the graph transition matrix to the power of $L$, scaling the eigenvalues to $\lambda^L$. Consequently, principal eigenvalues $\lambda \approx 1$ representing global cluster consensus are perfectly preserved, while smaller eigenvalues $\lambda < 1$ representing local multi-view noise decay exponentially to zero, explicitly filtering out high-frequency noise. This spectral decay mechanism strictly bounds the optimal choice of hierarchy depth $L$. An excessively large $L$ triggers an over-smoothing effect commonly observed in deep graph networks, where informative signals degrade and distinct cluster boundaries blur. Driven by this theoretical boundary, our experiments consistently demonstrate that setting $L \in [3,4]$ achieves an optimal balance. This range provides sufficient depth for effective noise filtration $\lambda^4 \to 0$ while remaining shallow enough to avoid over-smoothing. We will incorporate the complete spectral proof and this depth-selection guidance into the final version.
>
> **W2 (Optimization Complexity):** While introducing additional layers involves updating more variables, the practical optimization complexity does not escalate significantly. Because the number of anchors decreases progressively at each subsequent layer from $10k$ to $k$, the matrix dimensions for optimization in deeper layers become extremely small, where $c_l \ll n$. Furthermore, Figures 3 and 7 show that clustering performance saturates at $L=4$, avoiding the need for excessively deep architectures. To explicitly address this concern, we will include a per-layer empirical runtime analysis in the revised appendix to demonstrate the minimal computational overhead introduced by deeper layers.
>
> **W3 (Deep Learning Baselines):** Our initial focus on anchor-based baselines was to ensure a fair comparison under the same linear-time $\mathcal{O}(n)$ CPU setting. We nevertheless agree that including recent deep multi-view clustering methods would make the evaluation more comprehensive. More importantly, existing evidence already suggests that HAG-MVC remains competitive even against recent deep baselines. For example, compared with **AF-UMC**$^{[1]}$  (NeurIPS 2025), our method achieves higher ACC on **HW** (**0.9318** vs. **0.9035**) and **YTF** (**0.2858** vs. **0.1625**). Compared with **Coper**$^{[2]}$ (ICLR 2025), our method also obtains better ACC on **Reuters** (**0.6035** vs. **0.5315**). We will include these references and add more recent deep baselines in the revised manuscript to provide a more comprehensive comparison.
>
> **Q2 (Sensitivity to Initialization):** While we utilize $k$-means for initialization to accelerate convergence, HAG-MVC is significantly less sensitive to it compared to traditional anchor-based methods. This robustness stems directly from our core contribution: the multi-level co-evolution mechanism. Unlike traditional methods that keep anchors rigidly fixed after $k$-means, HAG-MVC continuously and dynamically updates the anchors. To explicitly demonstrate this, we will add an ablation study in the revised appendix comparing K-means initialization against simple random initialization. Preliminary tests confirm that while random initialization requires slightly more iterations to converge, it still achieves highly competitive clustering results thanks to the adaptive learning process.
>
> **Q3 (Framework Extensibility and Interpretability):** The proposed hierarchical anchor framework is not restricted to multi-view clustering and possesses high potential for extension to other graph-based learning tasks. The fundamental strength lies in its transparent and multi-level graph representation. Unlike deep neural networks that learn hidden features in opaque latent spaces, HAG-MVC constructs an explicit chain of bipartite graphs. This structure offers a mathematically grounded representation where the first layer captures local sample-to-anchor relations and higher layers progressively abstract higher-order semantic structures. Such an explicit hierarchy provides a traceable decision path from raw samples to global clusters, enhancing theoretical interpretability for any downstream task that requires modeling both local structure and high-level semantics.
>
> [1]Sun et al., AF-UMC: An Alignment-Free Fusion Framework for Unaligned Multi-View Clustering, NeurIPS 2025;
>
> [2]Eisenberg et al., Coper: Correlation-based Permutations for Multi-View Clustering, ICLR 2025.

---

> > ### Author Rebuttal · Reviewer_TmHk · 2026-04-06
> >
> > The author completely addressed my concerns.

---

> > > ### Author Response · Authors · 2026-04-07
> > >
> > > Thank you for your acknowledgment and for your careful reading of our rebuttal. We are glad that our response addressed your concerns, and we will incorporate the relevant revisions into the final version.

---

### Decision · Program_Chairs · 2026-04-30

**Decision:**

Accept (regular)

**Comment:**

This paper proposes a novel hierarchical anchor graph learning framework (HAG-MVC) for multi-view clustering. The framework introduces a multi-level pyramid of anchors with a novel co-evolution mechanism that jointly refines sample-to-anchor and anchor-to-anchor relations through alternating optimization, while maintaining linear computational complexity and preserving interpretability by keeping anchors in the original feature space. Extensive experiments on twelve diverse benchmark datasets demonstrate consistent and strong performance improvements over state-of-the-art anchor-based baselines, with comprehensive ablation studies and convergence analysis validating the design choices. After the rebuttal, all reviewers unanimously recommend acceptance for the paper. Therefore,  based on their recommendations, the paper is recommended for acceptance.